# TEST-TIME EFFICIENT PRETRAINED MODEL PORTFOLIOS FOR TIME SERIES FORECASTING

**Mert Kayaalp**[1]*, **Caner Türkmen**[2], **Oleksandr Shchur**[2], **Pedro Mercado**[2], **Abdul Fatir Ansari**[2], **Michael Bohlke-Schneider**[2], **Bernie Wang**[2]

[1] Dalle Molle Institute for Artificial Intelligence (IDSIA) - USI/SUPSI, Switzerland

[2] Amazon Web Services

## ABSTRACT

Is bigger always better for time series foundation models? With the question in mind, we explore an alternative to training a single, large monolithic model: building a portfolio of smaller, pretrained forecasting models. By applying ensembling or model selection over these portfolios, we achieve competitive performance on large-scale benchmarks using much fewer parameters. We explore strategies for designing such portfolios and find that collections of specialist models consistently outperform portfolios of independently trained generalists. Remarkably, we demonstrate that post-training a base model is a compute-effective approach for creating sufficiently diverse specialists, and provide evidences that ensembling and model selection are more compute-efficient than test-time fine-tuning.

## 1 INTRODUCTION

The dominant paradigm in pretrained time series models is the "bigger is better" view, supported by evidence that larger models improve forecast accuracy. Building on this observation, recent work has focused on scaling up both model size and training data to build better zero-shot forecasters (Das et al., 2024; Woo et al., 2024; Ansari et al., 2024a; Shi et al., 2024a; Edwards et al., 2025). However, these large, monolithic models come with high training and inference costs, which limits their practical use. To offset the need for ever-larger models, a promising alternative has emerged: allocating extra compute at test time. In NLP and computer vision, this includes generating multiple outputs for a single input and aggregating them—either by sampling from a single model (Sun et al., 2024) or using multiple models (Mavromatis et al., 2024). These examples, which partially inspire our work, belong to a broader class of methods that continue to achieve increasingly successful results across domains. Nevertheless, such strategies have not yet found their way to time series forecasting.

Inspired by these developments, we explore an alternative path. Instead of scaling up a single model, we propose to build a portfolio of smaller pretrained time series models and combine them at test time. This strategy offers greater flexibility and efficiency, but introduces two key challenges. First, how can we train a diverse set of models without incurring the full cost of training each one from scratch? Second, how should we combine the models at inference time to make accurate and efficient predictions? We address both challenges and show that this approach can match or even outperform larger monolithic models, while reducing inference costs and enabling modular design. More specifically, we make the following contributions.

1. We show that a portfolio of small pretrained models (based on encoder-decoder Chronos-Bolt Ansari et al. (2024b) architecture), each *specializing* on a subset of the training corpus, can match the accuracy of large monolithic models, while being more compute-efficient. By combining models at test time using ensembling or model selection, our approach achieves accuracy comparable to state-of-the-art pretrained forecasters, while dramatically reducing inference cost. We further observe that portfolios follow similar compute–performance scaling trends as monolithic models.

---

*Work done during an internship at AWS.

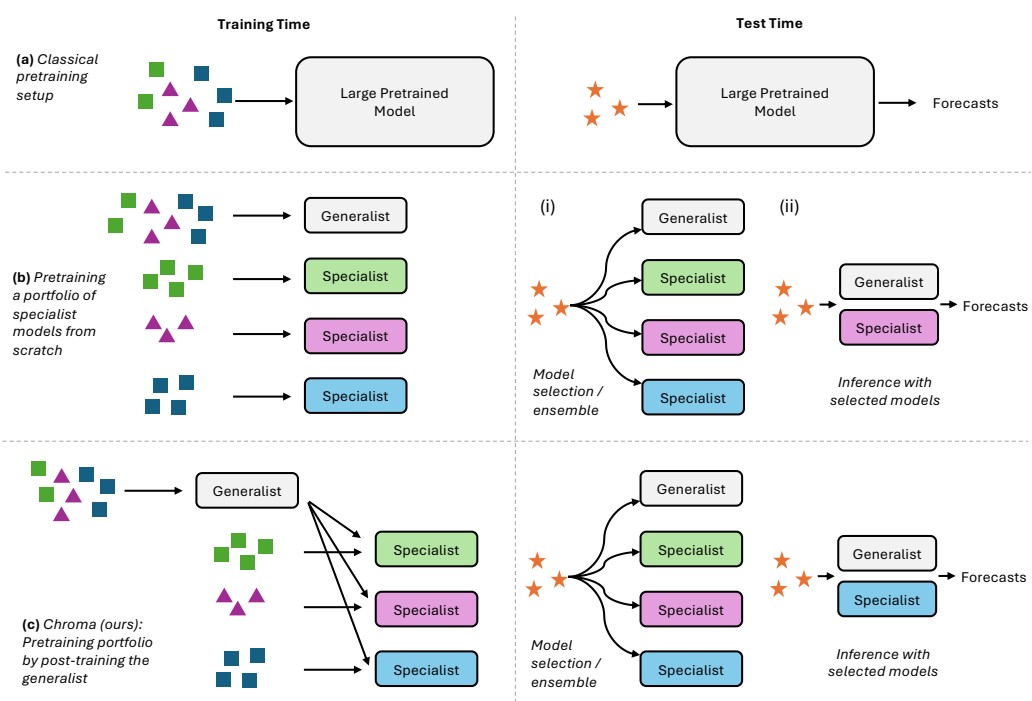

Figure 1: A diagram depicting the approaches explored in this work. (a) Typical setup for pretraining, where a single large model is trained on a large corpus composed of many different datasets. At test time, the model can be invoked *zero-shot* as well as with finetuning at test-time and used to obtain forecasts. (b) Building a model portfolio by training an array of smaller models, where each *specialist* is trained on a single smaller corpus representing a different modality or domain. At test time, these models are combined via fitting an ensemble or by selecting the best model, and the selected combination (or, single model) is used for forecasting. (c) Specialists can also be trained in two-stages, by *post-training* the generalist model. We demonstrate that this approach for inducing diversity into the model portfolio results in comparable accuracy, while reducing the training-time compute by an order of magnitude. The approach also leads to the same forecast accuracy as (a), in return for a much smaller number of total parameters used actively for inference.

2. We introduce an efficient strategy for constructing diverse model portfolios without training each model from scratch. The approach starts by pretraining a *generalist* model on the full data distribution, followed by targeted fine-tuning on data partitions defined by metadata such as frequency and domain, obtaining *specialists* (see, Figure 1). This reduces overall training time by 10x and yields a portfolio that maintains diversity and delivers accurate forecasts.

3. We explore different methods of combining pretrained portfolios at test time. Unsurprisingly, we find that the performance of our portfolios depends on the ability to quickly identify the best sparse combination among them in a new test task. We find that performing model selection or greedy ensemble selection among specialists is an efficient approach when compared to other methods of utilizing compute at test time (e.g., with fine tuning).

## 2 BACKGROUND AND RELATED WORK

### 2.1 TIME SERIES FORECASTING

The objective of time series forecasting is to predict the future $H$ values of a time series $x$, given the previous $C$ observations. Formally, we consider a time series $x = [x_1, \ldots, x_C]$ where $x_t \in \mathbb{R}$ denotes the value of the time series $x$ at time step $t$. The goal of probabilistic forecasting is to approximate the distribution of future time series values $x_{C+1:C+H}$ given the history $x_{1:C}$ with a

model $p_\theta$

$$p_\theta(x_{C+1:C+H}|x_{1:C}) \approx p(x_{C+1:C+H}|x_{1:C}).$$

Forecasting approaches fall into three broad categories (Benidis et al., 2022; Januschowski et al., 2020). *Local* models, such as ETS and ARIMA (Hyndman & Athanasopoulos, 2018), train separate models, often fully determined by several parameters ($\sim$10), for each individual time series. *Global* models, including deep learning methods (Salinas et al., 2020; Rangapuram et al., 2018; Oreshkin et al., 2020; Nie et al., 2023), learn shared parameters ($\sim$100K) across a single dataset of multiple related time series (e.g., energy consumption of different households). More recently, *pretrained* models such as Chronos (Ansari et al., 2024a), TimesFM (Das et al., 2024), and Moirai (Woo et al., 2024) have adapted ideas from large language models to time series forecasting. These models are trained on large and diverse collections of time series data and aim to generalize to new forecasting tasks without task-specific fine-tuning. With parameter counts typically exceeding 10M, they serve as universal forecasters capable of making accurate zero-shot predictions across a wide range of domains. To reduce inference costs associated with such large models, Ekambaram et al. (2024) proposed Tiny Time Mixers (TTMs)—lightweight models with around 1M parameters, specialized for specific context and forecast settings. Our approach shares the same goal of reducing the test-time cost, but instead of specializing a single small model per setting, we train a *portfolio* of small models and select or combine them at test time.

## 2.2 MODEL AND FORECAST COMBINATION

Traditional model combination methods, such as bagging (Breiman, 1996a) and boosting (Freund & Schapire, 1997), improve prediction accuracy by aggregating the outputs of multiple complementary models. Similar ideas have been applied in time series forecasting under the name *forecast combination* (Wang et al., 2023b). Forecasts are often combined using weighted averages of individual model predictions. These weights are typically shared across all time series in the dataset and are either set uniformly or in proportion to each model's validation performance (Pawlikowski & Chorowska, 2020). More flexible strategies learn these weights directly from data, for example using greedy ensemble selection (Caruana et al., 2004; Shchur et al., 2023).

In the context of pretrained models, some works have explored combining multiple LLMs, such as Mavromatis et al. (2024), which studies independently trained base models and their ensembles at test time. Here, the authors compare methods including weighted ensembling, uniform ensembling, and model selection. While their work is closely related to ours in the context of foundation model portfolios, it focuses exclusively on language tasks. In contrast, we focus on probabilistic time series forecasting, where the use of model combination and more generally test-time computation remains largely unexplored. Furthermore, beyond comparing model combination strategies as in Mavromatis et al. (2024), we also investigate how to build a diverse portfolio of test-time efficient models.

Another related approach for pretrained models is the mixture-of-experts (MoE) framework (Jacobs et al., 1991; Shazeer et al., 2017), where component models and combination rules are jointly trained. This approach is shown to be effective in time-series forecasting tasks as well (Shi et al., 2024b; Liu et al., 2024). Although these works showed that MoE can help reducing inference time and cost, it limits flexibility. In contrast, our approach trains the component models in the portfolio independently, and allows the use of different fusion strategies at test-time. Notably, this flexibility enables the fusion rule to be adapted according to available computational resources during inference. Additionally, our approach enhances interpretability during the prediction by clarifying the contribution of each portfolio member, whose training procedures are transparent.

## 3 CHROMA: A PORTFOLIO OF SMALL PRETRAINED FORECASTERS

In this section, we introduce our method for building portfolios of small pretrained forecasting models. Our approach consists of two stages. In the training stage, our goal is to build a diverse collection of small forecasting models. At test time, we combine their predictions to maximize accuracy for the given forecasting task. Applying this methodology to the publicly available pretrained models Chronos-Bolt (Ansari et al., 2024b), we construct our model portfolio, Chroma, which we describe in detail below.

### 3.1 Model architecture

We focus on portfolios composed of models with the same architecture, each pretrained independently on a different subset of the target distribution. The key difference is that we pretrain the full portfolio once, ahead of time, and reuse it across tasks—just as one would with a single large pretrained model.

We build Chroma using the training setup and architectures of Chronos-Bolt (Ansari et al., 2024b), a publicly available pretrained time series model based on the T5 encoder-decoder architecture (Raffel et al., 2020). Both pretraining and fine-tuning use average quantile loss, consistent with the original Chronos-Bolt implementation. Chronos-Bolt generates multi-step-ahead quantile forecasts and is trained on a diverse collection of time series datasets. We select this model due to its strong benchmark performance, open-source training pipeline and data, and support for a range of model sizes (e.g., `tiny`, `mini`), which enables flexible experimentation under different compute budgets. We train multiple versions of the model on different training datasets to induce diversity. We focus on smaller models ranging from 1M to 9M parameters, corresponding roughly to the `tiny (9m)` and below scale. Further implementation and training details are provided in Appendix B.

### 3.2 Building diverse portfolios

We now turn to the question of how to construct model portfolios that can be effectively combined at test time. It is well established that ensembles benefit from diversity among their members, provided each model is reasonably accurate (Zhou, 2012). However, diversity and individual model accuracy often trade off against each other (Wood et al., 2023).

In the context of pretrained models, diversity must be introduced during training, as weights are typically frozen at test time. The standard Chronos-Bolt training strategy aggregates all available time series datasets into a single, large corpus and trains one generalist model to perform well on average. In contrast, we aim to exploit structure in the data by explicitly training *specialist* (see, (Hinton et al., 2015)) models on disjoint subsets of the training corpus. Note that these models are not trained *at once* as in mixtures-of-experts or on the same data distribution but with perturbed samples as in bagging. Instead, we train each specialist model independently on a distinct partition of the corpus representing a different subproblem within the broader target distribution.

Specifically, we partition the data along metadata dimensions that reflect distinct characteristics of the time series. We focus on two such dimensions: frequency and application domain. Each specialist model is trained on a smaller subset of data defined by one of these attributes. For example, a domain-based portfolio may include an "energy" specialist trained only on energy-related datasets, while a frequency-based portfolio may contain an "hourly" specialist trained solely on hourly-resolution data. In each portfolio, we also include a *generalist* model of identical size trained on the full corpus for comparison. The training corpus of Chroma includes Chronos training data sets, plus some new ones for underrepresented frequencies and domains. The partitions and statistics of the datasets used are specified in Appendix C.

### 3.3 Reducing training cost through post-training

While forming portfolios, training all specialists from scratch on large datasets can result in significant computational cost. To mitigate this, we introduce diversity into the portfolio through *post-training*: we first train a generalist model of a given size, then fine-tune it briefly on different subsets of the training data to produce specialist models. This avoids the overhead of training each specialist model from random initialization.

This strategy is particularly important in our setup, where some data partitions may contain far fewer examples than others—for example, datasets with yearly resolution typically have far fewer observations than those with minutely resolution. In such cases, training from scratch may be inefficient or impractical. By leveraging post-training on a shared base model, we reduce overall training time and compute requirements while still matching the accuracy of portfolios trained independently. The benefits of post-training become even more critical in domains with larger models. While the largest time series models today are around 1 billion parameters, NLP models can reach up to 1 trillion. In such regimes, training each specialist from scratch is often infeasible, making post-training a practical and scalable alternative.

### 3.4 FORECAST COMBINATION AT TEST-TIME

Given a new time series dataset that was not seen by the portfolio members during training, we can utilize the models through one of two strategies: model selection or ensembling. In the first approach, we simply select the single best-performing model based on its accuracy on a validation set. In the second approach, we combine the predictions from multiple models using weighted averaging $\hat{y}_{\text{ens}} = \sum_{m=1}^{M} w_m \cdot \hat{y}_m$, where the weights are determined using the ensemble selection algorithm (Caruana et al., 2004). For completeness, we provide the full description of the ensemble selection algorithm in Appendix D. Both strategies depend on validation sets produced through time series cross-validation (Hyndman & Athanasopoulos, 2018) to rank or weight the predictions in the model portfolio.

## 4 EXPERIMENTS AND RESULTS

**Setup.** We extend the range of Chronos-Bolt model sizes to smaller variants and train portfolios of model sizes `1m`, `2m`, `4m`, and `tiny (9m)`. Details of these model architectures are provided in Appendix B. We extend the training corpus of Chronos[1] and partition this new corpus across two dimensions: frequency and domain.

For each model size, we first train a *generalist* using the original Chronos-Bolt corpus and a 200K-gradient-step training procedure following the reference implementations Ansari et al. (2024b), and incorporating mixup and synthetic data as specified in Ansari et al. (2024a). We then construct a portfolio of *specialists* by further fine-tuning (i.e., post-training) the generalist weights on different subsets of the training corpus for 1K gradient steps. In exploratory experiments, we also tested 3K, 10K, and 20K post-training steps, but observed no significant performance differences. As a result, we did not tune this variable further or include it as an experimental factor, and we use 1K gradient steps in our main experiments. This post-training phase amounts to only 0.5% of the generalist's training time, making our method of constructing specialists extremely lightweight. Each portfolio includes the generalist and its associated specialists.

To evaluate a portfolio on a target task, we hold out the last $H$ observations from the training set to create a validation window. All models in the portfolio generate forecasts for this window. We then either select the model with the lowest loss on the validation set, or fit a weighted ensemble using the greedy ensemble selection algorithm (Caruana et al., 2004) implemented in AutoGluon (Shchur et al., 2023). These are based on rolling-window evaluations (also referred to as backtesting or out-of-time folds) and can be performed even with a single time series. It can also be performed by increasing the number of evaluation windows. Namely, instead of using a single window for selecting which specialists provide the best forecasts, one could rely on multiple. The selected model or ensemble is then applied to the test set to produce final forecasts. Note that we use the same methodology throughout all experiments.

Most of our results are reported on Chronos Benchmark II[2] (BM2). Importantly, all datasets in the benchmark are zero-shot examples for our models (these datasets, including their training partitions, were not used in training of Chroma portfolios). Following the benchmark, we report the probabilistic forecasting results using the weighted quantile loss (WQL) and point forecasting results using the mean absolute scaled error (MASE). Finally, we also evaluate the larger Chroma variants on GIFT-Eval[3] (Aksu et al., 2024). For both benchmarks, we report aggregate performance using the geometric mean of relative errors. Specifically, for each dataset, we divide the model's error by that of a baseline (Seasonal Naive unless stated otherwise), and then compute the geometric mean across datasets (Fleming & Wallace, 1986). Further details on the experiment setup are given in Appendix B.

### 4.1 PERFORMANCE AND TEST-TIME EFFICIENCY

**Benchmark performance compared to the state of the art.** We report overall results for Chroma on BM2 in Figure 2 (left), comparing the performance of `4m` and `tiny (9m)` Chroma portfolios of frequency specialists to the state of the art in pretrained time series models. We find that, despite

---

[1] `https://huggingface.co/datasets/autogluon/chronos_datasets`
[2] `https://huggingface.co/spaces/autogluon/fev-leaderboard`
[3] `https://huggingface.co/spaces/Salesforce/GIFT-Eval`

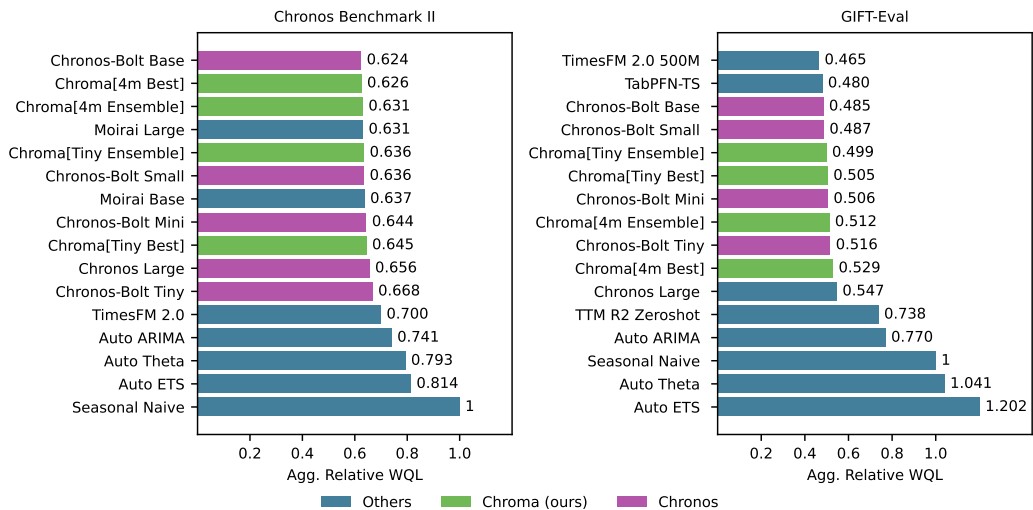

Figure 2: Results on Chronos Benchmark II (BM2) and GIFT-Eval. Results reported are for probabilistic forecasting, with weighted quantile losses (WQL), scaled relative to Seasonal Naive model, aggregated across all data sets using geometric mean. For Chroma, `Best` refers to performing model selection on a validation set while `Ensemble` refers to the ensemble selection algorithm.

having as little as 4M active parameters at test time, the Chroma portfolio under best model selection performs comparably to much larger monoliths such as Moirai-1.1 Large (311M parameters) (Woo et al., 2024), TimesFM-2.0 (500M parameters) (Das et al., 2024) or Chronos-Bolt Base (205M parameters).

We also report results on the GIFT-Eval benchmark, in Figure 2 (right), including only pretrained architectures and statistical baselines for comparison. In rolling window evaluation tasks, we perform model selection or ensembling on the Chroma model portfolio on the first evaluation window alone, and infer with this fixed selection across the subsequent windows. That is, the ensemble was fixed in the training window of the first rolling window, and was not refit in subsequent windows. In this benchmark, too, Chroma's aggregated probabilistic forecasting performance over 97 tasks is close to some much larger pretrained architectures, including TabPFN-TS (Hoo et al., 2025) which leverages significant test-time computation for in-context learning with a tabular foundation model. Note that we report results for TTM (Ekambaram et al., 2024) in zero-shot mode only. Moreover, in order not to misrepresent the performance of TTM, which specializes in high-frequency time series, we omit TTM's results on BM2 which primarily contains low-frequency time series datasets. Further details on benchmark evaluations, including point forecasting performance, are given in Appendix G.

**Portfolio design and combination method.** We now investigate the impact of key design choices in portfolio construction and test-time combination across different model sizes. To better isolate the effect of these choices, we report results on BM2 using a different reference point: instead of comparing to the Seasonal Naive baseline, we compute relative errors with respect to a single generalist model of size `1m`. We also include a similarly sized portfolio of generalists (models trained with the same large corpus of data but with different random seeds).

We observe that building time series model portfolios by partitioning datasets by frequency is preferable to partitioning them by application domain, as this results in $\sim 5\%$ decrease in overall MASE and $\sim 4\%$ in WQL, across all model sizes considered (Table 1). This is consistent with earlier work such as TTM (Ekambaram et al., 2024) who used frequency to partition datasets. When comparing the use of ensemble vs. model selection, although ensemble selection leads to a slight improvement across model sizes and metrics, we find the evidence for this is not as strong as our previous conclusion. Therefore, while using ensembling to combine Chroma models can result in improvements in accuracy, this difference may not be strong enough to justify the added computational costs of using multiple models for inference.

Table 1: Aggregated relative WQL and MASE on BM2, scaled against a single generalist model of size `1m`. The best two models for each metric and model size are given in bold.

|  |  | WQL | | | | MASE | | | |
|---|---|---|---|---|---|---|---|---|---|
|  |  | 1m | 2m | 4m | tiny | 1m | 2m | 4m | tiny |
| **Domain** | **Ensemble (Greedy)** | 0.957 | 0.936 | 0.932 | 0.915 | 0.964 | 0.932 | 0.928 | 0.918 |
|  | **Model Selection** | 0.963 | 0.950 | 0.939 | 0.923 | 0.973 | 0.948 | 0.933 | 0.930 |
| **Frequency** | **Ensemble (Greedy)** | **0.926** | **0.898** | **0.890** | **0.896** | **0.910** | **0.884** | **0.878** | **0.887** |
|  | **Model Selection** | **0.918** | **0.916** | **0.880** | **0.909** | **0.920** | **0.899** | **0.887** | **0.893** |
| **Generalists** | **Ensemble (Greedy)** | 0.987 | 0.951 | 0.939 | 0.919 | 0.974 | 0.944 | 0.931 | 0.928 |
|  | **Model Selection** | 0.990 | 0.966 | 0.942 | 0.944 | 0.979 | 0.961 | 0.935 | 0.941 |
| **Single Generalist** | **None** | 1.000 | 0.977 | 0.960 | 0.958 | 1.000 | 0.971 | 0.962 | 0.961 |

Finally, we observe that simply combining multiple generalists trained using the same large data set does not result in significant performance gains over just using a single generalist. We hypothesize this is due to variance being a negligible factor in the test error committed by models pretrained on large corpora—an observation that can be supported by Lin et al. (2024), who argues that current LLMs remain underfitted owing to the relatively small number of training epochs. We discuss this observation further in Appendix F. Specifically, we show that in the regime where Chroma models are trained, bias component of the generalization error dominates the error due to variance. Since specialists reduce bias within a specific part of the overall data distribution, intelligent model combination or ensembling at test-time can reduce the overall bias of the portfolio. This is as opposed to the traditional ensembling approaches like bagging that tries to reduce error by decreasing variance.

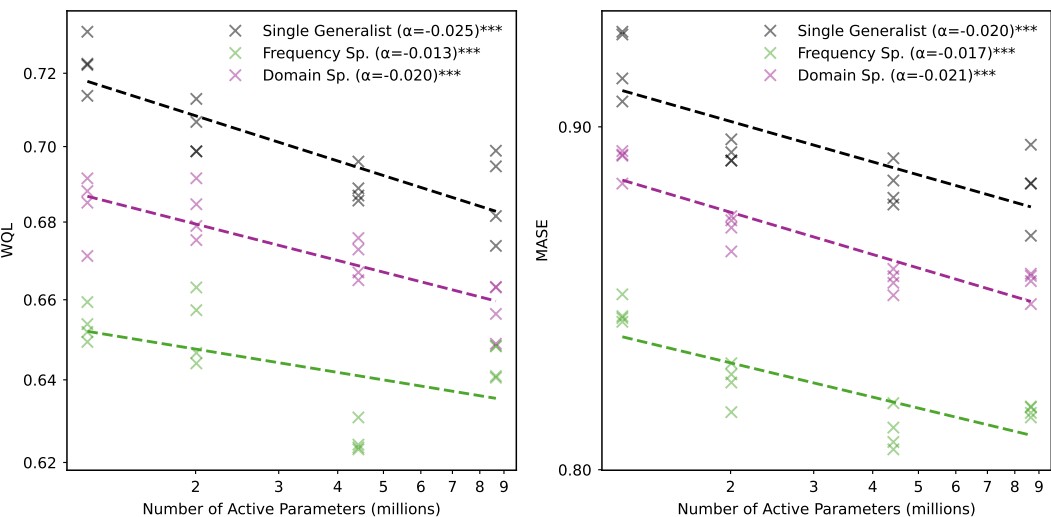

Figure 3: Scaling behavior of Chroma portfolios. Results are presented for performing model selection from a Chroma portfolio of domain or frequency specialists. Each individual run is for an independently trained generalist in the portfolio, with four trials reported per experiment setting. $\alpha$ refers to the slope of the scaling fit, and asterisks denote statistical significance of the fitted coefficient at the 5% level ($p < 0.05$). Reported results are aggregated across BM2.

**Scaling behavior.** Having observed that our conclusions hold across different model sizes, we turn our attention to quantifying how Chroma *scales*. Similar to recent works (Hoffmann et al., 2022; Kaplan et al., 2020; Shi et al., 2024a; Edwards et al., 2025), we carry out ordinary least squares fits to the test error of Chroma (on BM2) and the number of *active* parameters at inference time. We present results for model selection on the Chroma portfolio in Figure 3. We observe that despite individually training on much smaller datasets, the overall model portfolio's performance scales at rates comparable to the individual generalist models. In further experiments, we found that our training setup of post-training on generalists to obtain the portfolio is essential to this conclusion

(see, Appendix G). We therefore conclude that, provided datasets can be proportionally scaled, our approach can be scaled up to larger model sizes to obtain even better performance.

**Test-time compute efficiency.** Chroma delivers strong forecast accuracy, but at the cost of additional test-time computation. Since we must either select the best model from the portfolio or combine multiple models via greedy ensembling, each model in the portfolio must run inference on both the validation and test sets. This raises a natural question: how does this compute overhead compare to alternative strategies for adapting pretrained models at test time?

The simplest option is to use a single generalist model in a zero-shot manner—no adaptation, and only one forward pass. Our approach sits in the middle: it avoids fine-tuning but requires multiple inference passes and some lightweight model selection or combination logic. At the high end of the compute spectrum is test-time fine-tuning, where a single generalist model is adapted to the target task using gradient updates.

To quantify these trade-offs, we compare all three strategies—zero-shot generalist, Chroma (with model selection or ensembling), and fine-tuned generalist—by measuring total test-time floating point operations (FLOPs) and reporting accuracy across the board. Fine-tuning is done with 1K gradient steps for reference. This lets us position Chroma in terms of both accuracy and compute efficiency relative to common adaptation strategies. Our results are presented in Figure 4, where different model sizes and combination methods are shown along the efficient frontier of accuracy vs. the total test-time compute required. Fine-tuning a generalist model for 1K gradient steps yields strong accuracy but requires nearly 10x more compute than simple forward passes. Chroma offers a middle ground: it achieves competitive accuracy using a fixed portfolio of small models, with only lightweight model selection or ensembling at test time. This places it near the center of the efficiency frontier, offering a favorable trade-off between accuracy and computational cost. Further fine-tuning may improve accuracy, but it shifts the operating point toward significantly higher compute requirements.

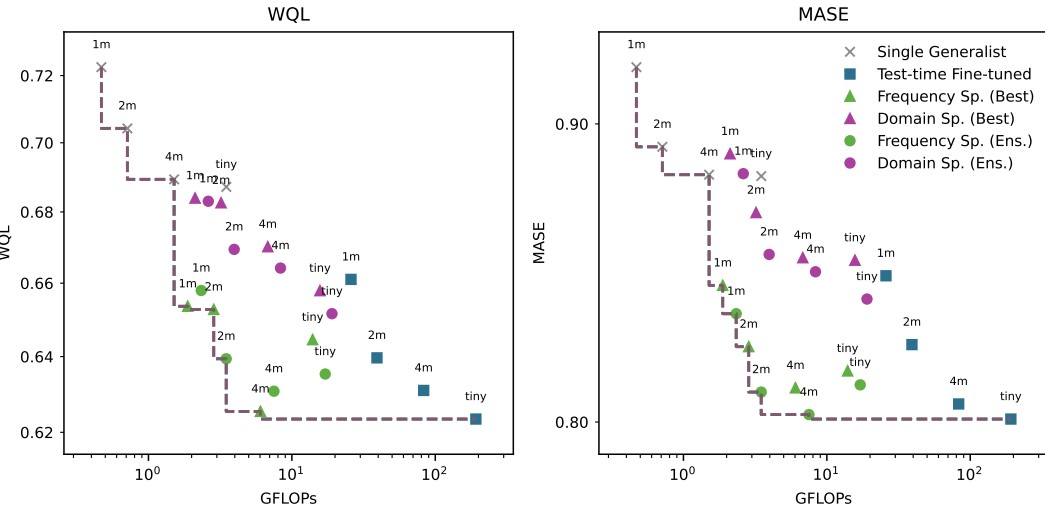

Figure 4: Efficient frontier of Chroma portfolios, compared to using a single generalist model without any test-time computation (i.e., zero-shot, ×) and fine-tuning a generalist model (■). Test-time computation is computed as the estimated FLOPs for performing model fitting, fine tuning, or model selection in addition to a single forward pass for inference, per time series in the test set. Reported results are aggregated across BM2.

## 4.2 FURTHER EXPERIMENTS

**Ablation studies.** We perform three ablation studies summarized in Table 2. A key element in the portfolios we consider is the inclusion of a generalist (see, e.g., Hinton et al. (2015)). We estimate the effect of excluding the generalist on MASE and WQL errors averaged across different model sizes, specialist types (domain vs. freq) and datasets. In both cases, we find average error increases of $\sim 2\%$, while these regressions in performance are not measured to be statistically significant.

Table 2: Summary results of ablation studies. Results are given in the estimated increase in normalized error (where seasonal naive error is 1) on BM2 for the ablation performed, and the p-value associated with the t-test of the regression coefficient.

|  | MASE | | WQL | |
| --- | --- | --- | --- | --- |
|  | **Increase in Error** | **p-value** | **Increase in Error** | **p-value** |
| **No Generalists in Portfolio** | 0.016 | 0.107 | 0.017 | 0.118 |
| **Ensemble - Performance-weighted** | 0.186 | 0.000 | 0.142 | 0.000 |
| **Ensemble - Simple Average** | 0.242 | 0.000 | 0.180 | 0.000 |
| **Specialists Trained From Scratch** | 0.002 | 0.798 | -0.001 | 0.918 |

Second, we observe that the benefit of specialist portfolios comes primarily from the diversity they introduce. However, this diversity is only useful if we combine models intelligently at test time. In particular, our ensembles do not work simply by reducing variance through averaging, as might be expected in settings where randomness in training dominates model differences. We confirm this in ablation studies: both simple averaging and performance-weighted averaging (cf. Pawlikowski & Chorowska, 2020) lead to significant performance drops when applied to our pretrained portfolios. This contrasts with much of the prior forecasting literature, where ensembles of task-specific models are often effective even with naive combination strategies (Wang et al., 2023b).

An essential part of our methodology in training a diverse portfolio of models is to pretrain a single generalist before inducing diversity by post-training with different data subsets. In our final ablation study, we compare this to training all specialists from random initializations (see, Figure 1 (b)). On average, this does not lead to any regressions in performance. However, in scaling studies we find that post-trained models scale better (see, Appendix G).

**Credit assignment and interpretability of the portfolio.**    We also investigate which specialists are activated during ensembling or model selection for a given test task. Figure 5 illustrates this activation pattern. The heatmaps are structured as matrix plots, where rows represent tasks and columns represent specialists (categorized by domain or frequency). We observe that all specialists are activated for some tasks, indicating no "mode collapse" where only a single model dominates. Notably, when the test task shares a domain or frequency with a particular specialist, that specialist tends to be highly activated. This suggests a practical efficiency gain: in many real-world applications, the nature of the test-time task is known in advance. This knowledge can be leveraged to manually select relevant specialists, thereby reducing test-time computational cost. Additionally, Figure 5 also offers a layer of interpretability. For each task, the specialist weights provide meaningful insights that can aid practitioners in understanding the model's behavior.

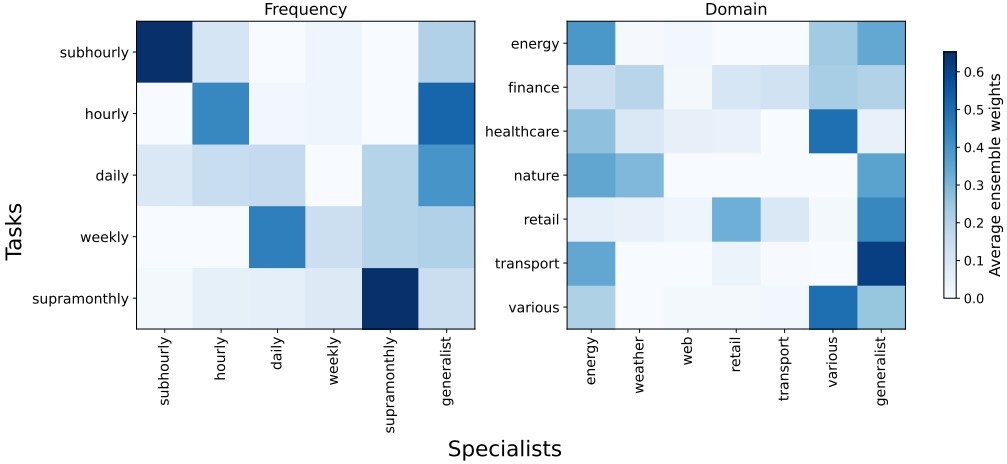

Figure 5: Distribution of ensemble weights for 4m specialist portfolios across distinct tasks, grouped with respect to the their domain or frequency information.

## 5 CONCLUSION

In this work, we introduced Chroma: a portfolio of small, pretrained time series forecasters. Chroma consists of specialist models, each fine-tuned on disjoint subsets of the training data from a generalist pretrained model. When model selection or ensembling is applied at test time, Chroma achieves competitive performance on forecasting benchmarks from Ansari et al. (2024a) and Aksu et al. (2024), matching the accuracy of much larger monolithic models. Our experiments demonstrate the test-time efficiency of Chroma, as well as its scalability. We believe that the methodology behind Chroma opens a promising new direction for enhancing the accuracy of pretrained forecasting models, especially under limited compute budgets. Furthermore, while our focus is on time series forecasting, the underlying principles may also extend to other domains such as natural language processing and computer vision. Our strategy offers an alternative to approaches like Best-of-$N$ which samples from a single base model. Instead, it proposes an inference-aware portfolio formation strategy: train a single generalist base model, fine-tune it into a portfolio of specialized experts, and apply model selection at test time. Moreover, we focused on test-time efficiency because inference often dominates the computational budget in deployed forecasting systems.

**Limitations and future work.** Although we did not study the impact of the pretraining phase in detail, our design choices follow standard practice: T5-architecture based Chronos-Bolt model with straightforward modifications, see Appendix B. Also, our theoretical observations in Appendix F provides a bias-variance perspective that helps explain the observed gains and suggests implications for the role of pretraining. Understanding how the number of training iterations and other pretraining hyperparameters interact with our conclusions would be a valuable direction for future research. Similarly, our experiments were conducted with encoder-decoder transformer architectures. Future work can investigate portfolios of transformers based on other types of architectures. Moreover, while we hand-crafted the specialist portfolios using known dataset features (e.g., domain or frequency), future work could explore more principled approaches. For instance, foundation model specialists can be formed by incorporating boosting-style objectives (Freund & Schapire, 1997). Finally, this work focused on weighted ensembling; future research could explore more sophisticated methods such as stacked ensembling (Breiman, 1996b) that can lead to higher accuracy.

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

# APPENDIX

## A  ADDITIONAL LITERATURE

Prior work shows that ensembles of neural networks outperform single networks and provide better uncertainty quantification. Specifically, it has been reported that training the same model architecture from different random initializations is an effective strategy for sufficiently diverse portfolios Lee et al. (2015), and deep ensembles utilize this idea Lakshminarayanan et al. (2017); Fort et al. (2019); Zaidi et al. (2021). At the foundation-model scale, however, such work is less common. Notably, Sun et al. (2022) builds LLM ensembles by varying random seeds, while Wang et al. (2023a) uses low-rank adapters for resource efficiency. We adopt a similar approach while forming our generalist portfolios and likewise observe gains over single models. However, we further show that larger improvements can be achieved with specialist portfolios, where models are fine-tuned on targeted subsets of the data distribution. As discussed in Appendix F, in the foundation-model regime, variance reduction from ensembling is less decisive than improved coverage of heterogeneous data.

In time-series context, we also find that partitioning by frequency to induce specialization across portfolio members performs particularly well. This aligns with a broader line of work that exploits frequency-based representation and basis expansions for time-series forecasting. Namely, Darlow et al. (2024) proposes a foundation model that predicts coefficients of sinusoidal bases for functional decomposition, and Kim et al. (2024) represents time series directly in the frequency space. As opposed to those methods aiming to learn a single, monolithic model, our methodology is focused on specialization, i.e., training multiple experts on frequency-based clusters and aggregating their predictions. In classical statistical forecasting, beyond our baselines such as Auto-ETS and Auto-ARIMA, there are also recent methods such as FFORMA (Montero-Manso et al., 2020), which learns to predict ensemble weights over a pool of univariate forecasters using time-series features, and SCUM (Petropoulos & Svetunkov, 2020), which aggregates established statistical methods via median-based fusion.

## B  DETAILS ON MODEL ARCHITECTURES AND HYPERPARAMETERS

We adopt Chronos-Bolt as the architecture for Chroma (Ansari et al., 2024b). Chronos-Bolt is based on the T5 encoder-decoder architecture (Raffel et al., 2020). It chunks the historical time series context into patches of multiple observations, as in PatchTST (Nie et al., 2023), which are then input into the encoder. The decoder then uses the representations generated by the encoder to directly generate quantile forecasts across multiple future steps—a method known as direct multi-step forecasting. This differs from the original Chronos (Ansari et al., 2024a) models that rely on autoregressive decoding. The Chronos-Bolt architecture following a standard pattern — namely, patching the inputs, using transformer blocks, and multi-step-ahead quantile prediction — makes it a representative choice for time series foundation models, as this pattern is similar to other recent SOTA time-series models.

We also adopt the training methodology of Chronos-Bolt, which was trained on nearly 100B time series observations from varying frequencies and application domains. The training objective is the mean weighted quantile loss (WQL) over 9 equally spaced quantiles between $0.1$ and $0.9$. For simplicity, we keep the same training corpus as Chronos-Bolt when training our generalist models. When training specialists, we added new datasets to the training corpus to balance data for underrepresented frequency and domain classes (see, Appendix C. Our generalist models are not trained on these new datasets to remain consistent with the Chronos-Bolt setup. Although we do not directly compare generalists and their corresponding specialists in our main experiments, we also explored training generalists on the full specialist dataset. We have observed that this had little effect. The performance of individual generalists, test-time fine-tuned generalists, and generalist portfolios considered in the main paper remained largely unchanged.

Note that the models we train are smaller in size ranging from 1 million parameters to 9 million parameters. The 9 million parameter model architecture exists in the released Chronos-Bolt and T5-efficient line-ups as the `tiny` model size.[4] The rest of the models were shrunk, roughly keeping

---

[4]https://huggingface.co/google/t5-efficient-tiny

the same *aspect ratio* as the Chronos models. These hyperparameter choices are given in Table 3. Importantly, we set the patch size and patch stride parameters to 16. Across all models, the number of heads is set to 4, the dropout rate to $0.1$, decay the learning rate using a triangular schedule initialized at $10^{-3}$, and use a batch size of 256. We use the AdamW optimizer (Loshchilov & Hutter, 2019) with weight decay set to $0.1$, gradient clipping at $1.0$ and $\beta_1 = 0.98, \beta_2 = 0.9$. All models are pretrained for 200K iterations, as outlined in (Ansari et al., 2024a). For post-training fine-tuning, as discussed in the main paper, we perform 1K gradient update steps. For a portfolio of $N$ specialists, this yields a speedup of $200K \times N/(200K + 1K \times N) \approx 10x$ for typical portfolio sizes. This speedup applies equally to wall-clock time, FLOPs, and GPU-hours.

Table 3: Architectural hyperparameters for the model sizes considered in the paper. $d_{\text{ff}}, d_{\text{kv}}$, and $d$ refer to the feedforward model hidden dimension, key-value dimension, and the model (embedding) dimension respectively. $L_e$ and $L_d$ denote the number of layers in the encoder and decoder.

|      | $d_{\text{ff}}$ | $d_{\text{kv}}$ | $d$ | $L_d$ | $L_e$ |
|------|------|------|------|------|------|
| tiny | 1024 | 64 | 256 | 4 | 4 |
| 4m   | 768  | 64 | 192 | 3 | 3 |
| 2m   | 640  | 32 | 160 | 2 | 2 |
| 1m   | 512  | 32 | 128 | 1 | 2 |

We trained and evaluated all models on NVIDIA A10G Tensor Core GPUs, used through AWS EC2 G5 family of instances. For reference, our `1m`, `2m`, `4m`, and `9m` (`tiny`) model sizes take $1.4$, $1.7$, $3.3$, and $4.8$ hours on average to train from scratch for 200K iterations, on a single A10G GPU. Post-training runs typically take several minutes.

## C    DETAILS ON SPECIALISTS

In order to train the models of Chroma portfolios, datasets inside the general training corpus[5] as well as additional datasets from Monash (Godahewa et al., 2021) and UCI (Asuncion et al., 2007) repositories, are partitioned in two ways, into six partitions of domain and five partitions of frequency. Details of these assignments, as well as some summary statistics of the data sets are provided in Table 4.

Models trained from scratch follow the exact same training procedure as the generalist models, including 200K training iterations, albeit with access to only a specific part of the training corpus.

For Chroma specialists, which were post-trained, we randomly selected a generalist model as the initial weights and fine-tuned each specialist via 1K gradient steps, using the same training setup as mentioned above. Due to computational constraints, we trained only one set of specialists for each model size and partitioning scheme, however during experiments, we evaluated each portfolio 5 times varying the generalist in the portfolio and reported their average. Similarly, for all other results reported on single generalists, we give an average of 5 independently trained models.

## D    DETAILS ON MODEL SELECTION

Ensemble selection algorithm Caruana et al. (2004) is a standard approach for forecast combination in both point and probabilistic forecasting (Deng et al., 2022; Shchur et al., 2023). Given a set of models producing predictions $\hat{y}_1, \ldots, \hat{y}_M$, where each $\hat{y}_m$ is the tensor of predictions for all item, time steps and quantile levels, the goal is to find optimal weights $w_1, \ldots, w_M$ for the ensemble prediction $\hat{y}_{\text{ens}} = \sum_{m=1}^{M} w_m \cdot \hat{y}_m$.

The ensemble selection algorithm optimizes validation loss by greedily adding models to an equally-weighted ensemble with replacement. Since an equally-weighted ensemble with replacement is equivalent to a weighted average with fractional weights, this can be interpreted as optimizing weights $w_1, \ldots, w_M$ via coordinate-wise ascent.

---

[5]https://huggingface.co/datasets/autogluon/chronos_datasets/

Table 4: Datasets used in training, summary statistics and and their assignments to domain and frequency partitions.

| | Nr. Time Series | Nr. Observations | Missing Rate | Avg. Length | Frequency | Domain |
|---|---|---|---|---|---|---|
| indian_power_generation | 13 | 13,858 | 0.0% | 1066 | daily | energy |
| wind_farms_daily | 337 | 119,549 | 16.8% | 355 | daily | energy |
| iowa_liquor_subset | 847 | 1,803,263 | 0.0% | 2129 | daily | retail |
| online_retail_I | 1,742 | 587,893 | 0.0% | 337 | daily | retail |
| online_retail_II | 4,772 | 2,369,710 | 0.0% | 497 | daily | retail |
| monash/covid_mobility | 450 | 180,379 | 13.0% | 401 | daily | transport |
| monash/covid_mobility_without_missing_values | 450 | 180,379 | 0.0% | 401 | daily | transport |
| uber_tlc_daily | 262 | 47,422 | 0.0% | 181 | daily | transport |
| m4_daily | 4,227 | 10,023,836 | 0.0% | 2371 | daily | various |
| monash/bitcoin | 18 | 82,458 | 9.8% | 4581 | daily | various |
| monash/bitcoin_without_missing_values | 18 | 75,364 | 0.0% | 4187 | daily | various |
| monash/us_births | 1 | 7,305 | 0.0% | 7305 | daily | various |
| monash/sunspot | 1 | 73,924 | 4.4% | 73924 | daily | weather |
| monash/sunspot_without_missing_values | 1 | 73,924 | 0.0% | 73924 | daily | weather |
| monash/temperature_rain | 422 | 305,950 | 0.5% | 725 | daily | weather |
| ushcn_daily | 1,218 | 47,080,115 | 7.7% | 38654 | daily | weather |
| monash/electricity_hourly | 321 | 8,443,584 | 0.0% | 26304 | hourly | energy |
| solar_1h | 5,166 | 45,254,160 | 0.0% | 8760 | hourly | energy |
| wind_farms_hourly | 337 | 2,869,414 | 17.0% | 8515 | hourly | energy |
| mexico_city_bikes | 494 | 38,687,004 | 0.0% | 78314 | hourly | transport |
| monash/pedestrian_counts | 66 | 3,132,346 | 0.0% | 47460 | hourly | transport |
| monash/rideshare | 156 | 84,396 | 44.2% | 541 | hourly | transport |
| taxi_1h | 2,428 | 1,794,292 | 0.0% | 739 | hourly | transport |
| uber_tlc_hourly | 262 | 1,138,128 | 0.0% | 4344 | hourly | transport |
| m4_hourly | 414 | 373,372 | 0.0% | 902 | hourly | various |
| monash/kdd_cup_2018 | 270 | 2,942,364 | 17.1% | 10898 | hourly | weather |
| monash/kdd_cup_2018_without_missing_values | 270 | 2,942,364 | 0.0% | 10898 | hourly | weather |
| electricity_15min | 370 | 41,936,458 | 0.0% | 113342 | subhour | energy |
| monash/aus_solar_1min | 1 | 493,149 | 0.0% | 493149 | subhour | energy |
| monash/aus_wind_1min | 1 | 493,144 | 0.0% | 493144 | subhour | energy |
| monash/elecdemand | 1 | 17,520 | 0.0% | 17520 | subhour | energy |
| monash/london_smart_meters | 5,560 | 166,528,896 | 0.0% | 29951 | subhour | energy |
| monash/solar_10_minutes | 137 | 7,200,720 | 0.0% | 52560 | subhour | energy |
| monash/solar_4_seconds | 1 | 7,397,222 | 0.0% | 7397222 | subhour | energy |
| solar | 5,166 | 543,049,920 | 0.0% | 105120 | subhour | energy |
| taxi_30min | 2,428 | 3,589,798 | 0.0% | 1478 | subhour | transport |
| heart_rate | 4 | 5,500 | 0.0% | 1375 | subhour | various |
| monash/wind_4_seconds | 1 | 7,397,147 | 0.0% | 7397147 | subhour | weather |
| m4_monthly | 48,000 | 11,246,411 | 0.0% | 234 | supra | various |
| monash/m3_other | 174 | 13,325 | 0.0% | 77 | supra | various |
| brazil_gas_prices | 187 | 128,471 | 0.0% | 687 | weekly | energy |
| monash/electricity_weekly | 321 | 50,076 | 0.0% | 156 | weekly | energy |
| monash/solar_weekly | 137 | 7,124 | 0.0% | 52 | weekly | energy |
| m4_weekly | 359 | 371,579 | 0.0% | 1035 | weekly | various |
| monash/kaggle_web_traffic_weekly | 145,063 | 16,537,182 | 0.0% | 114 | weekly | web |
| weatherbench_daily | 225,280 | 78,991,953,920 | 0.0% | 350639 | daily | weather |
| wiki_daily_100k | 100,000 | 274,100,000 | 0.0% | 2741 | daily | web |

**Algorithm**: Greedy Ensemble Weighting

1. Initialize weights $w^{(0)} = \mathbf{0} \in \mathbb{R}^M$.
2. For iterations $j = 1, \ldots, S$:
   (a) Select model that minimizes validation loss:
   $$m^{(j)} = \arg\min_m \ L(\hat{y}_{\text{ens}}, y_{\text{val}}),$$
   where $\hat{y}_{\text{ens}}$ uses the updated weights
   $$\frac{(j-1)w^{(j-1)} + e_m}{j},$$
   and $e_m$ is the $m$-th canonical basis vector.
   (b) Update weights:
   $$w^{(j)} = \frac{(j-1)w^{(j-1)} + e_{m^{(j)}}}{j}.$$
3. Return final weights $w^{(S)}$.

In our evaluation we use $S = 100$ steps of ensemble selection and use the last window of $H$ observations in the training portion of each series as the validation set $y_{\text{val}}$. As in the rest of the paper, we use the weighted quantile loss (WQL) as the loss function $L$.

# E  FLOPs COMPUTATION

In our test-time compute we use approximations to the total number of floating point operations used at test time. For this, we use approximations which we define here for completeness. As in Appendix B, let $L_e, L_d$ denote the number of layers in the encoder and decoder, and $d$ the model dimension respectively. We also denote $T$ as the *effective sequence length*, which is the number of tokens that are given to the encoder as the context. For our work, $T = \frac{2048}{16} = 128$ where 2048 is the original sequence length and 16 is the patch size, for all models considered. We compute

$$\text{FLOPs}_{\text{forward}} \approx (L_e + L_d) \times T \times d^2 \times \left(24 + \frac{4T}{d}\right).$$

When model fine-tuning using backward passes are required at test time, we calculate

$$\text{FLOPs}_{\text{train}} \approx 3 \times \text{FLOPs}_{\text{forward}},$$

since the backward pass usually incurs about twice the cost of the forward pass, plus some overhead.

Note we compute two numbers for test-time compute, namely, *total* test-time compute and *amortized* compute. By the former, we refer to the total effort of performing model selection or ensembling with a given portfolio of $N$ models, and performing one inference pass with the resulting selection. For example, total test time compute for performing model selection is given as $(N + 1) \times \text{FLOPs}_{\text{forward}}$ where the $N$ term results from the necessity to infer with each model once during selection, and the 1 term to the single inference pass with the selected model. Similarly, for ensembling, this factor is $\approx (N + 2.5)$ where 2.5 is the average number of models selected for the ensemble. Finally, when we report *amortized* test-time compute, we omit the $N$ factor as the initial cost of performing the model selection or fine-tuning will be amortized over all future inference passes using the resulting ensemble or model.

# F  BIAS–VARIANCE IN FOUNDATION MODEL PORTFOLIOS

As we have seen in Table 1, portfolios composed of generalist models, each trained on the full training distribution, yield only small performance improvements over a single generalist model. In contrast, specialist portfolios in Chroma, whether based on frequency or domain, offer substantial gains. This may seem counterintuitive, since each generalist model is trained on the entire training distribution rather than a specific subset, and on average, a model from the generalist portfolio outperforms a model from the specialist portfolio in a direct, head-to-head comparison. In this section, we offer an explanation for this observation using the bias–variance decomposition of the expected generalization error.

The generalization error of a model is known to be decomposable as (Vapnik, 1999)

$$\text{Average Test Error} = \text{Irreducible noise} + \text{Bias} + \text{Variance}.$$

Here, irreducible noise is inherent to the task and cannot be reduced. Typically, model combination methods target either the bias or variance components to lower the overall error.

In a generalist portfolio, all models are trained on the same training corpus and therefore share same biases. As a result, ensembles' overall bias remains unchanged from that of the individual models in the portfolio. However, such ensembles reduce variance. Indeed, if the models are independently sampled, the variance decreases by a factor of $N$, where $N$ is the number of models in the ensemble. Note that this setup is analogous to classical bagging in ensemble learning (Breiman, 1996a; Zhou, 2012), where multiple models trained on bootstrapped subsets of the full data set are averaged to reduce variance without affecting bias.

We argue that **in the regime where pretrained time series models are trained, generalization error due to bias dominates the error due to variance**. As a result, having a generalist portfolio does not result in significant performance gains over a single model.

To confirm this argument, we estimate the bias and variance components of the error of a single model. To do so, we train ten distinct generalists using different random seeds following the same training procedure outlined in Appendix B. In order to discard the "irreducible noise" component of

the error, we test our models on synthetic signal inputs without a noise component. Specifically, we draw 10 thousand time series from a mixture of periodic Gaussian process kernels and offset these with random trends and constants. On this dataset, we compute

$$\text{Bias}(x) = \left(\bar{f}(x) - y\right)^2 \qquad \text{Variance}(x) = \frac{1}{M} \sum_{i=1}^{M} \left(f_i(x) - \bar{f}(x)\right)^2,$$

where $f_i(x)$ denotes the point forecast produced by model realization $i$ for test input $x$, $\bar{f}(x)$ is the average forecast across all $M$ realizations, and $y$ represents the true value of the signal corresponding to $x$ over the prediction horizon. The results are provided in Table 5. Observe that over all model size spectrum we consider in this work, the bias component of the error dominates the variance component across all generalist models.

Table 5: Bias and variance components of the error. It can be seen that bias component dominates the variance component across all model sizes.

|  | Bias | Variance |
| --- | --- | --- |
| 1m | 65.305 | 10.067 |
| 2m | 49.031 | 12.298 |
| 4m | 22.009 | 6.423 |
| tiny | 20.063 | 8.367 |

Our observation for time-series foundation models aligns with the argument of Lin et al. (2024), which states that, in current large language models, the variance component of the error becomes strictly smaller than the bias because these models are typically trained for only one or a few epochs on their massive training corpora, leaving them underfit.

These observations also help us understand why Chroma works so well. While building pretrained model portfolios, Chroma trains specialist models by fine-tuning a generalist base model. Each of these specialists reduces bias within a specific part of the overall data distribution. As a result, a specialist portfolio consists of models with distinct biases.

Crucially, we leverage this diversity in the biases effectively through an intelligent model combination at test time. In contrast to traditional ensembling approaches, such as bagging, which reduce variance by averaging out randomness across models, Chroma's advantage does not rely on variance reduction. Instead, it operates through targeted bias reduction. For instance, when applying model selection, it selects the most appropriate specialist for each test input, and the overall bias of the ensemble effectively becomes equal to the minimum bias among the models in the portfolio.

Table 2 supports this perspective, as well. A simple, uniform averaging over specialist portfolios fail to deliver performance gains comparable to model selection or greedy ensembling. This finding stands in contrast to much of the forecast combination literature, where heterogeneous ensembles of task-specific models typically benefit from averaging. In our case, however, the models are used in a zero-shot manner, and require test time input specific selection to be effective. This is consistent with our earlier observation that, in the pretrained regime, generalization error is dominated by bias rather than variance. Since variance is already low, reducing bias through targeted specialist selection offers a more effective way of improving performance.

## G  ADDITIONAL EXPERIMENT RESULTS

This section complements Section 4 by providing additional results and plots.

### G.1  SCALING BEHAVIOR

As discussed in Section 4, we perform ordinary least squares fits on the test error of BM2 to analyze the scaling behavior of Chroma. Each run corresponds to an independently trained generalist within the portfolio, with four trials reported per experimental setting. The parameter $\alpha$ denotes the slope of

the scaling fit, and asterisks indicate statistical significance of the fitted coefficient at the 5% level (p < 0.05).

Figures 6 and 7 show the test error plotted against the number of active parameters at inference time for model selection and ensembling on the Chroma portfolio, respectively. Indeed, the overall portfolio performance scales at rates comparable to those of the individual generalist models.

In contrast, Figures 8 and 9 present results when the Chroma portfolio is trained from scratch, rather than post-training. Here, as mentioned in the main paper, the behavior differs significantly. Therefore, we conclude that post-training generalist models to form the portfolio is essential for Chroma's scaling behavior.

We also provide scaling plots in terms of amortized test-time compute in Figures 10, 11, 12, 13, calculated according to Appendix E, which provide the same conclusions.

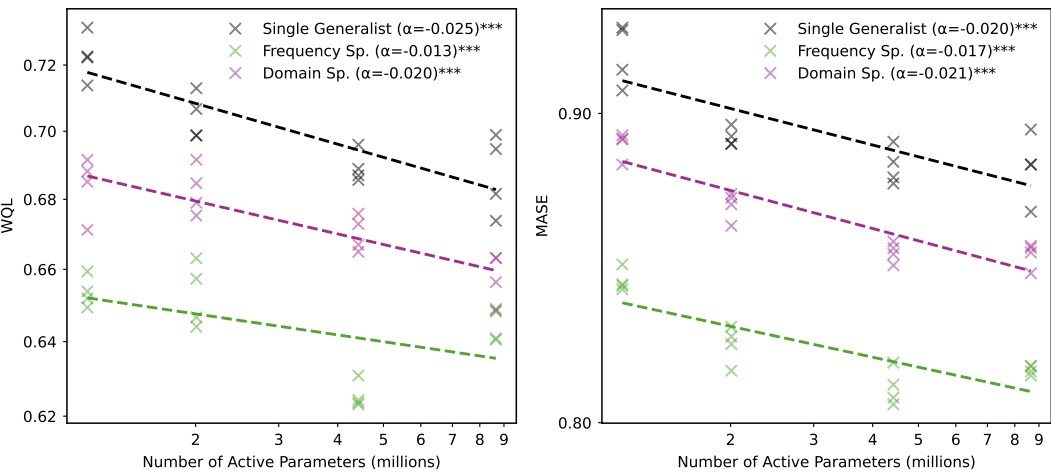

Figure 6: Scaling behavior of Chroma portfolios. Results are presented for performing model selection from a Chroma portfolio of domain or frequency specialists, where each of these specialists are formed with post-training.

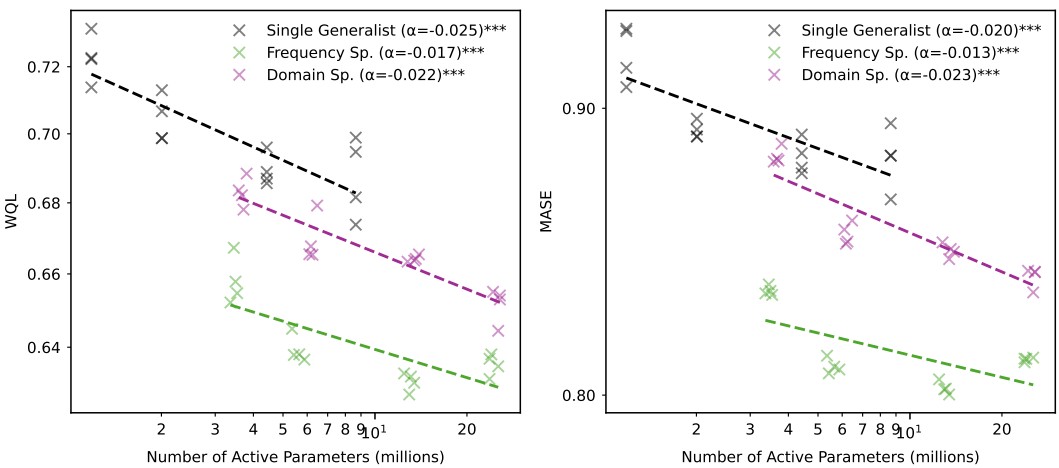

Figure 7: Scaling behavior of Chroma portfolios. Results are presented for performing ensembling on a Chroma portfolio of domain or frequency specialists, where each of these specialists are formed with post-training.

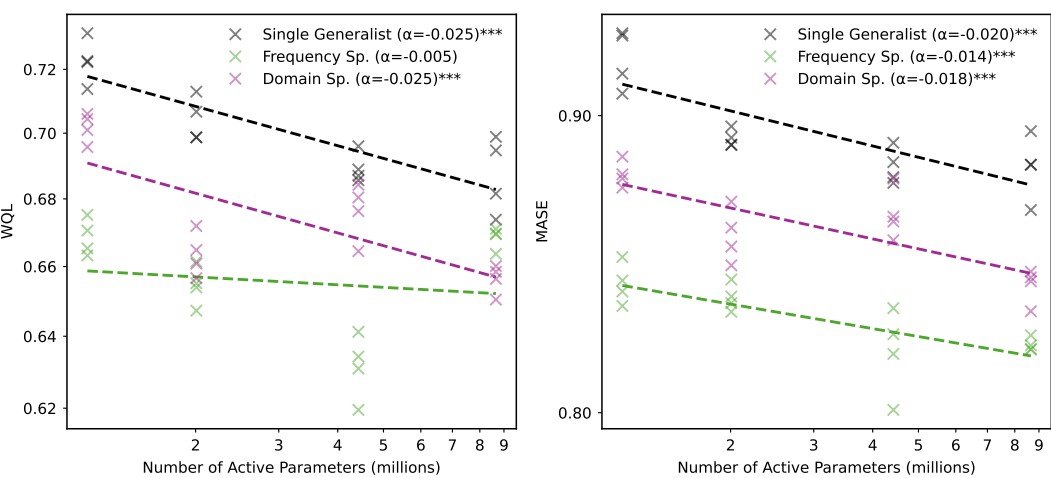

Figure 8: Scaling behavior of portfolios, when each of the specialists are trained from scratch. Results are presented for performing model selection from a portfolio of domain or frequency specialists.

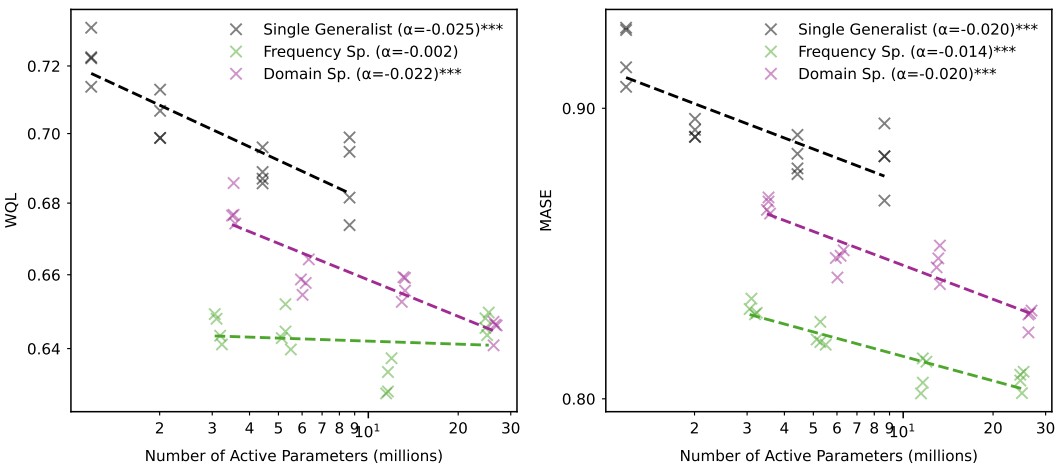

Figure 9: Scaling behavior of portfolios, when each of the specialists are trained from scratch. Results are presented for performing ensembling on a portfolio of domain or frequency specialists.

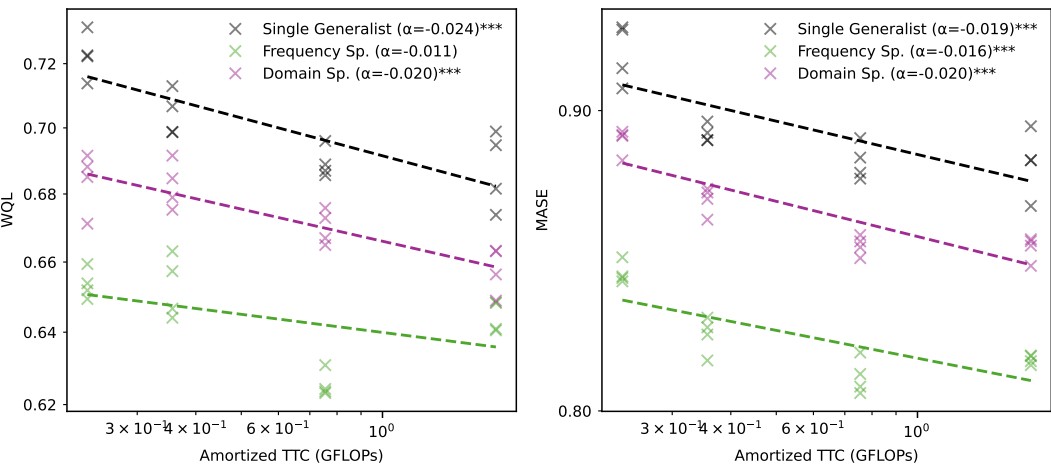

Figure 10: Scaling behavior of Chroma portfolios with respect to the amortized test time compute, when the specialists are formed with post-training and model selection is performed during test-time.

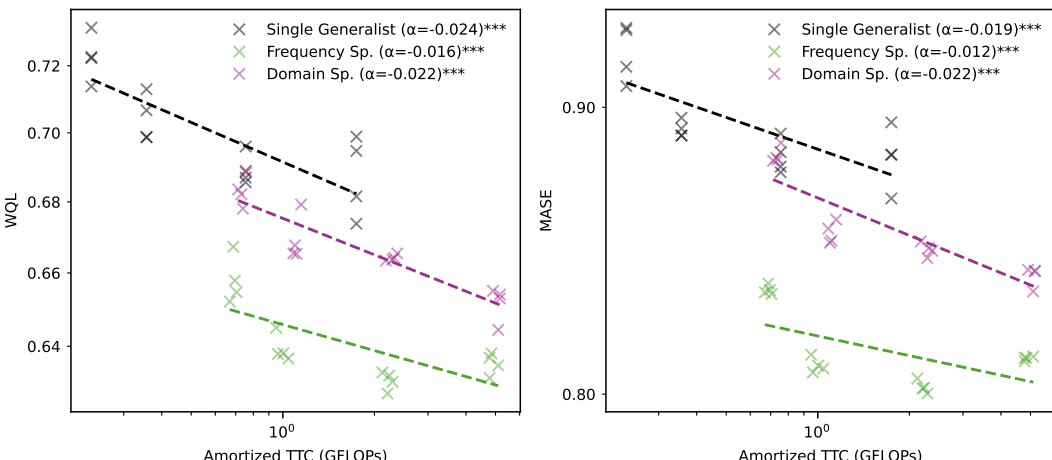

Figure 11: Scaling behavior of Chroma portfolios with respect to the amortized test time compute, when the specialists are formed with post-training and greedy ensembling is performed during test-time.

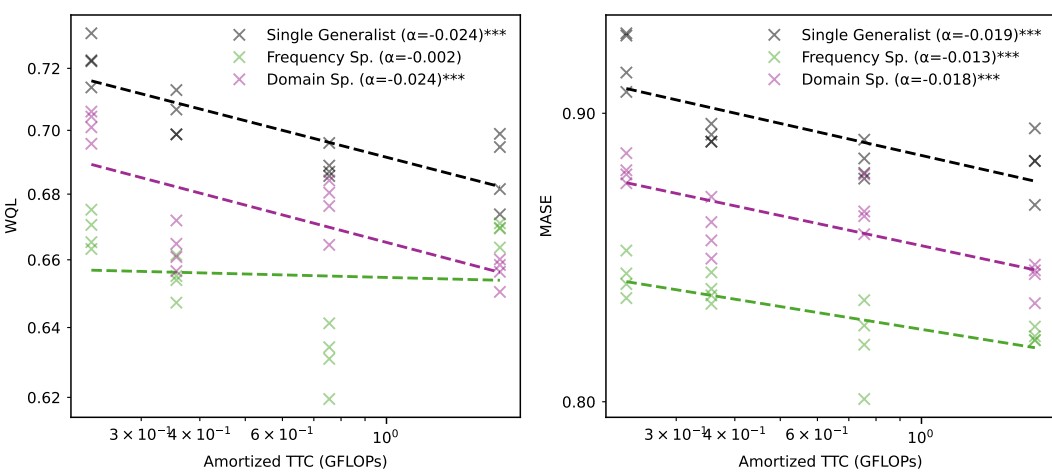

Figure 12: Scaling behavior of portfolios with respect to the amortized test time compute, when the specialists are trained from scratch and model selection is performed during test-time.

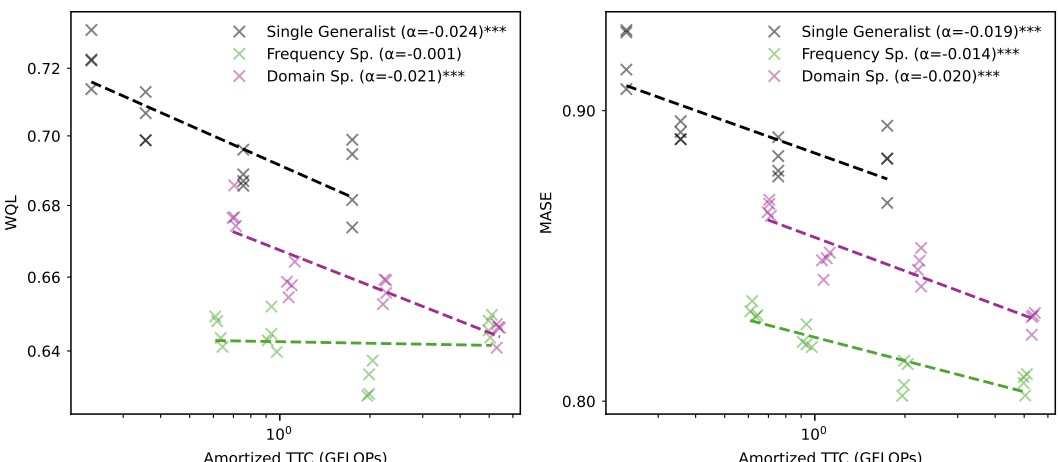

Figure 13: Scaling behavior of portfolios with respect to the amortized test time compute, when the specialists are trained from scratch and greedy ensembling is performed during test-time.

## G.2 TEST-TIME COMPUTE EFFICIENCY

Figures 14 and 15 demonstrate the tradeoff between total test-time compute (in terms of FLOPs) and performance. FLOPs are estimated as described in Appendix E, and incorporate the costs of model fitting, fine-tuning, or model selection, in addition to a single forward pass for inference per item in the test set.

Figure 14 presents results from Chroma portfolios formed through post-training, while Figure 15 shows portfolios composed of models trained from scratch. Both are compared against two baselines: a single generalist model evaluated in a zero-shot setting (i.e., without any test-time computation, ×), and a generalist model that has been fine-tuned (■). All results are aggregated over the BM2 benchmark.

The conclusion remains the same in both figures. Chroma portfolios with frequency specialists lie at the test-time compute-efficiency frontier, whether the specialists are formed with post-training or trained from scratch.

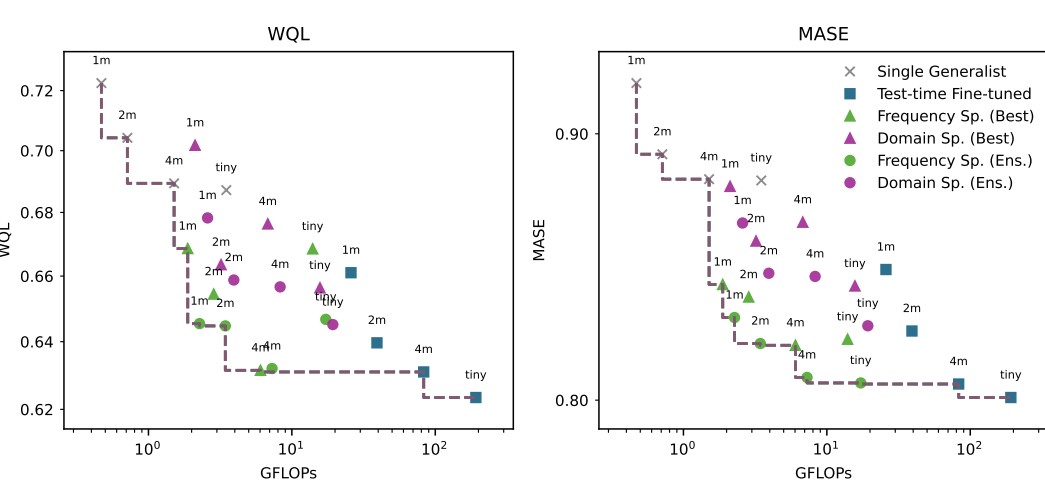

Figure 14: Total test-time compute (GFLOPs) and performance tradeoff, presented for Chroma portfolios built through post-training.

Figure 15: Total test-time compute (GFLOPs) and performance tradeoff, presented for portfolios when the specialist models are trained from scratch.

### G.3 CREDIT ASSIGNMENT AND INTERPRETABILITY OF THE PORTFOLIO

In Figures 16, 17, 18, and 19, we extend the analysis from Figure 5 by applying the same methodology to portfolios composed of model architectures `1m`, `2m`, `4m`, and `tiny`, as depicted in Appendix B. We examine which specialists are activated during ensembling for each test task. Like in Figure 5, the heatmaps are presented as matrix plots, with rows representing tasks and columns representing specialists (categorized by domain or frequency). Tasks are grouped according to their domain or frequency, and the weights are aggregated across tasks and normalized per group. We can see that our findings from Figure 5 remain consistent across portfolios of different model sizes.

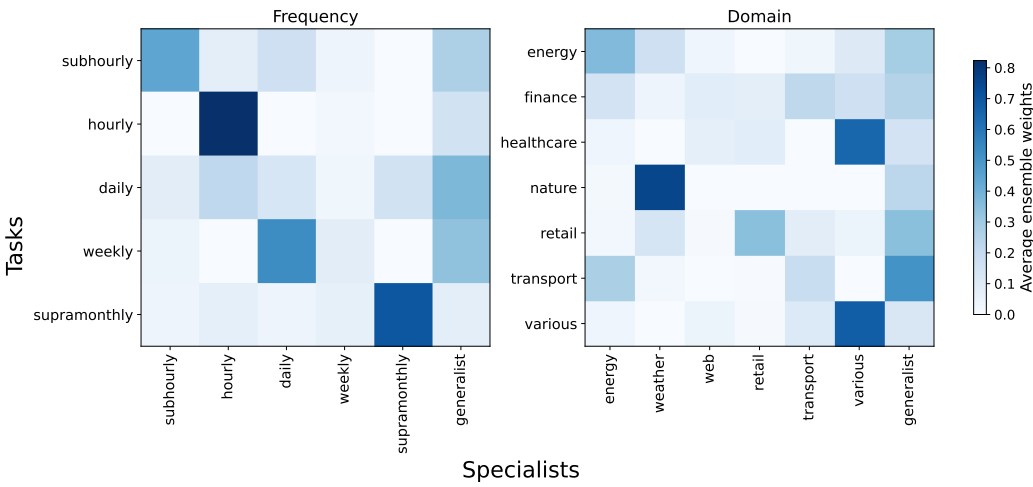

Figure 16: Distribution of ensemble weights for `1m` specialist portfolios across distinct tasks.

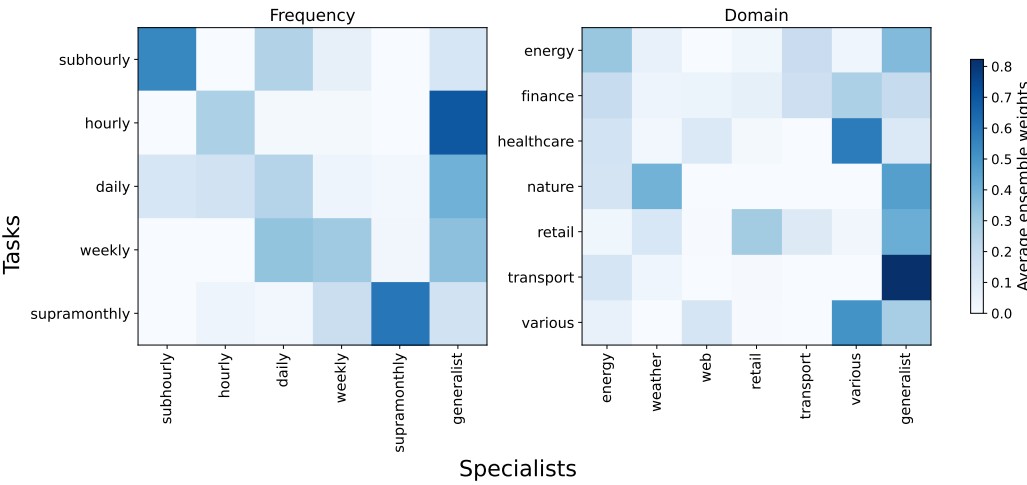

Figure 17: Distribution of ensemble weights for `2m` specialist portfolios across distinct tasks.

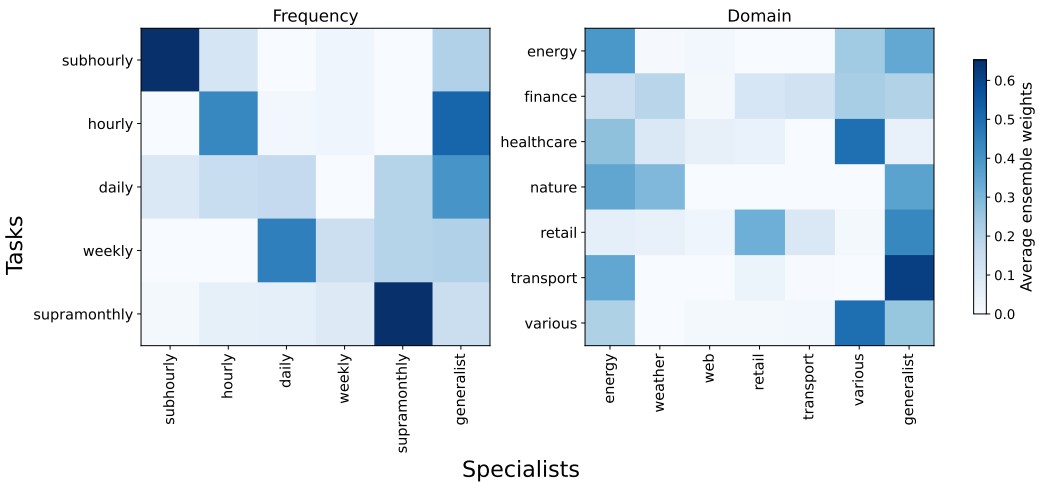

Figure 18: Distribution of ensemble weights for `4m` specialist portfolios across distinct tasks.

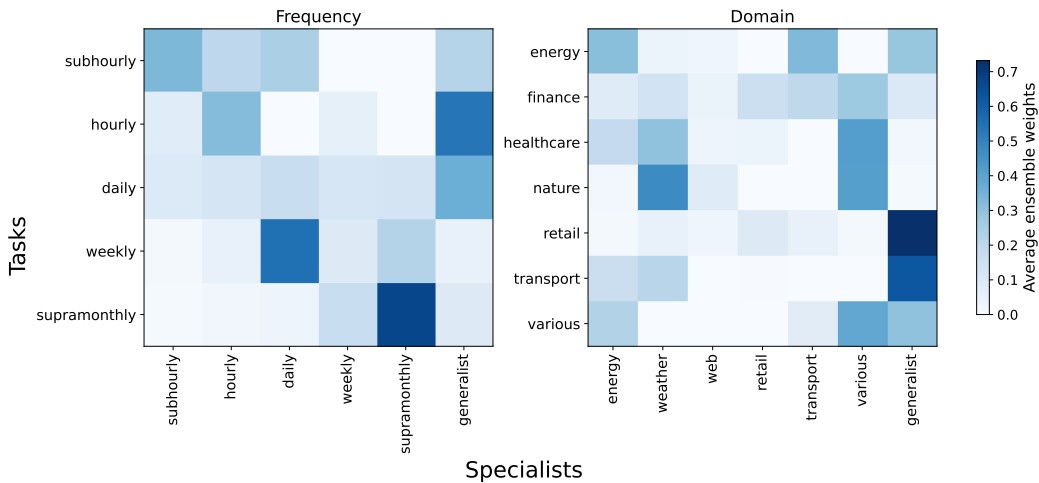

Figure 19: Distribution of ensemble weights for `tiny` specialist portfolios across distinct tasks.

## G.4 ABLATION STUDY

We perform three sets of ablations in this study, quantifying the effects of dropping generalists from ensembles, using simple averaging as the ensembling strategy, and for training Chroma portfolio models from scratch instead of post-training. For each study, we fit mixed linear models controlling for the evaluation dataset, and quantify the effect of the ablation and its statistical significance. While overall results for ablations are given in the main text, we also provide detailed results per portfolio type and model size as well.

Table 6: Ablation study, fixed effects and p-values for removing the generalist from model selection or ensembles for all model sizes and portfolio types.

|  | Group | WQL - Fixed Effect | WQL - p-value | MASE - Fixed Effect | MASE - p-value |
|---|---|---|---|---|---|
| **Portfolio Type** | domain | 0.025 | 0.000 | 0.024 | 0.000 |
| **Portfolio Type** | freq | 0.009 | 0.015 | 0.007 | 0.0536 |
| **Model Size** | 1m | 0.017 | 0.015 | 0.012 | 0.184 |
| **Model Size** | 2m | 0.017 | 0.015 | 0.018 | 0.0238 |
| **Model Size** | 4m | 0.018 | 0.01225 | 0.018 | 0.014 |
| **Model Size** | tiny | 0.016 | 0.0023 | 0.014 | 0.014 |
| **Overall** | N/A | 0.017 | 0.000 | 0.016 | 0.000 |

Table 7: Ablation study, fixed effects and p-values for replacing greedy weighted averages with simple and performance weighted ensembles.

|  | Group | Ensemble Type | WQL - Fixed Effect | WQL - p-value | MASE - Fixed Effect | MASE - p-value |
|---|---|---|---|---|---|---|
| **Portfolio Type** | domain | Perf | 0.149 | 0.000 | 0.193 | 0.000 |
| **Portfolio Type** | domain | Simple | 0.192 | 0.000 | 0.264 | 0.000 |
| **Portfolio Type** | freq | Perf | 0.136 | 0.000 | 0.179 | 0.000 |
| **Portfolio Type** | freq | Simple | 0.167 | 0.000 | 0.221 | 0.000 |
| **Model Size** | 1m | Perf | 0.142 | 0.000 | 0.179 | 0.000 |
| **Model Size** | 1m | Simple | 0.177 | 0.000 | 0.231 | 0.000 |
| **Model Size** | 2m | Perf | 0.137 | 0.000 | 0.179 | 0.000 |
| **Model Size** | 2m | Simple | 0.172 | 0.000 | 0.232 | 0.000 |
| **Model Size** | 4m | Perf | 0.152 | 0.000 | 0.199 | 0.000 |
| **Model Size** | 4m | Simple | 0.196 | 0.000 | 0.265 | 0.000 |
| **Model Size** | tiny | Perf | 0.138 | 0.000 | 0.186 | 0.000 |
| **Model Size** | tiny | Simple | 0.174 | 0.000 | 0.241 | 0.000 |
| **Overall** | N/A | Perf | 0.142 | 0.000 | 0.186 | 0.000 |
| **Overall** | N/A | Simple | 0.180 | 0.000 | 0.242 | 0.000 |

Table 8: Ablation study, fixed effects and p-values for replacing post-training with training each specialist from scratch.

|  | Group | WQL - Fixed Effect | WQL - p-value | MASE - Fixed Effect | MASE - p-value |
|---|---|---|---|---|---|
| **Portfolio Type** | domain | -0.008 | 0.08575 | -0.005 | 0.20533 |
| **Portfolio Type** | freq | 0.007 | 0.08575 | 0.008 | 0.014 |
| **Model Size** | 1m | -0.002 | 0.791 | -0.003 | 0.554 |
| **Model Size** | 2m | -0.009 | 0.08575 | 0.004 | 0.4732 |
| **Model Size** | 4m | -0.001 | 0.791 | 0.009 | 0.1925 |
| **Model Size** | tiny | 0.010 | 0.08575 | -0.004 | 0.392 |
| **Overall** | N/A | -0.001 | 0.791 | 0.002 | 0.554 |

### G.4.1 PORTFOLIOS TRAINED FROM SCRATCH

In the result Table 2 of our ablation study, we show that there is no statistically significant difference in accuracy between portfolios trained from scratch and portfolios trained with post-training. In Table 9, we support this result by a rework of Table 1 that also includes portfolios of models that are trained from scratch, as opposed to post-training.

Table 9: Aggregated relative WQL and MASE on BM2 including portfolios trained from scratch, scaled against a single generalist model of size `1m`.

| | | WQL | | | | MASE | | | |
|---|---|---|---|---|---|---|---|---|---|
| | | 1m | 2m | 4m | tiny | 1m | 2m | 4m | tiny |
| **Domain** | **Ensemble (Greedy)** | 0.957 | 0.936 | 0.932 | 0.915 | 0.964 | 0.932 | 0.928 | 0.918 |
| | **Model Selection** | 0.963 | 0.950 | 0.939 | 0.923 | 0.973 | 0.948 | 0.933 | 0.930 |
| **Frequency** | **Ensemble (Greedy)** | 0.926 | 0.898 | 0.890 | 0.896 | 0.910 | 0.884 | 0.878 | 0.887 |
| | **Model Selection** | 0.918 | 0.916 | 0.880 | 0.909 | 0.920 | 0.899 | 0.887 | 0.893 |
| **Domain (scratch)** | **Ensemble (Greedy)** | 0.953 | 0.927 | 0.923 | 0.911 | 0.947 | 0.924 | 0.925 | 0.905 |
| | **Model Selection** | 0.987 | 0.928 | 0.943 | 0.919 | 0.968 | 0.936 | 0.942 | 0.919 |
| **Frequency (scratch)** | **Ensemble (Greedy)** | 0.907 | 0.905 | 0.893 | 0.910 | 0.906 | 0.895 | 0.882 | 0.880 |
| | **Model Selection** | 0.935 | 0.914 | 0.885 | 0.940 | 0.920 | 0.913 | 0.895 | 0.896 |
| **Generalists** | **Ensemble (Greedy)** | 0.987 | 0.951 | 0.939 | 0.919 | 0.974 | 0.944 | 0.931 | 0.928 |
| | **Model Selection** | 0.990 | 0.966 | 0.942 | 0.944 | 0.979 | 0.961 | 0.935 | 0.941 |
| **Single Generalist** | **None** | 1.000 | 0.977 | 0.960 | 0.958 | 1.000 | 0.971 | 0.962 | 0.961 |

It can be seen that there is no significant difference between these two type of portfolios in terms of accuracy. Similarly, from the individual tasks point of view, we also were not able draw a direct conclusion about how from-scratch and post-trained models compare. Despite this performance similarity, the portfolios trained from scratch required 200x more compute for each specialist.

### G.5 DETAILED BENCHMARK RESULTS

While we provide the aggregated results in the main text, in this section, we provide detailed results for each individual dataset included in the benchmarks. Specifically, for BM2, we report the results for WQL and MASE in Tables 10 and 11, respectively. For GIFT-Eval, WQL results are shown in Tables 12 and 13, while MASE results are presented in Tables 14 and 15.

## H USE OF LLMS

Large language models are only used for a few word choices in the text. They are not used for the implementation of this research.

Table 10: Weighted quantile losses (WQL) on Chronos Benchmark II

| | Chroma (4m, freq, best) | Chroma (4m, freq, ens.) | Chroma (tiny, freq, best) | Chroma (tiny, freq, ens.) | Chronos Bolt Base | Chronos Bolt Mini | Chronos Large | Moirai Large | TimesFM 2.0 | Auto ARIMA | Auto ETS | Seasonal Naive |
|---|---|---|---|---|---|---|---|---|---|---|---|---|
| **australian_electricity** | 0.038 | 0.037 | 0.043 | 0.042 | **0.036** | 0.045 | 0.080 | **0.035** | 0.067 | 0.067 | 0.191 | 0.084 |
| **car_parts** | 1.010 | 1.013 | 1.005 | 1.000 | **0.995** | 1.004 | 1.054 | **0.975** | 1.652 | 1.333 | 1.340 | 1.600 |
| **cif_2016** | 0.017 | 0.017 | 0.016 | 0.017 | 0.016 | 0.018 | **0.014** | **0.014** | 0.053 | 0.033 | 0.039 | 0.015 |
| **covid_deaths** | 0.056 | 0.058 | 0.056 | 0.052 | 0.047 | 0.061 | 0.058 | 0.038 | 0.215 | **0.029** | **0.032** | 0.133 |
| **dominick** | 0.361 | 0.362 | 0.360 | 0.363 | 0.345 | 0.350 | **0.330** | **0.332** | 0.371 | 0.485 | 0.484 | 0.453 |
| **ercot** | 0.017 | 0.017 | 0.018 | 0.018 | 0.021 | 0.024 | **0.017** | **0.017** | 0.021 | 0.041 | 0.118 | 0.037 |
| **etth** | **0.075** | **0.075** | 0.079 | 0.079 | **0.071** | 0.077 | 0.076 | 0.090 | 0.085 | 0.089 | 0.132 | 0.122 |
| **ettm** | 0.062 | 0.059 | 0.072 | 0.067 | 0.052 | **0.051** | 0.071 | 0.073 | **0.052** | 0.105 | 0.085 | 0.141 |
| **exchange_rate** | **0.009** | 0.010 | 0.011 | 0.011 | 0.012 | **0.009** | 0.012 | 0.012 | 0.015 | 0.011 | 0.010 | 0.013 |
| **fred_md** | 0.036 | 0.047 | 0.048 | 0.042 | 0.042 | 0.033 | **0.027** | 0.052 | **0.027** | 0.035 | 0.033 | 0.122 |
| **hospital** | 0.054 | 0.054 | 0.054 | 0.054 | 0.057 | 0.058 | 0.055 | **0.053** | **0.050** | 0.059 | 0.053 | 0.073 |
| **m1_monthly** | 0.142 | 0.140 | 0.140 | 0.139 | 0.139 | 0.133 | **0.130** | 0.167 | **0.130** | 0.154 | 0.165 | 0.191 |
| **m1_quarterly** | **0.081** | 0.085 | **0.083** | **0.083** | 0.101 | 0.098 | 0.103 | 0.110 | 0.113 | 0.088 | 0.085 | 0.150 |
| **m1_yearly** | 0.133 | 0.133 | 0.161 | 0.161 | 0.151 | 0.162 | 0.185 | **0.126** | 0.145 | **0.133** | 0.139 | 0.209 |
| **m3_monthly** | 0.094 | 0.094 | 0.093 | **0.093** | 0.093 | 0.095 | 0.095 | 0.097 | **0.089** | 0.098 | 0.093 | 0.149 |
| **m3_quarterly** | 0.078 | 0.075 | 0.072 | 0.072 | 0.076 | 0.078 | 0.073 | **0.070** | 0.075 | 0.077 | **0.070** | 0.101 |
| **m3_yearly** | **0.126** | **0.124** | 0.148 | **0.075** | 0.129 | 0.157 | 0.147 | 0.131 | 0.144 | 0.156 | 0.129 | 0.167 |
| **m4_quarterly** | 0.079 | 0.077 | 0.076 | 0.075 | 0.077 | 0.078 | 0.082 | 0.076 | **0.062** | 0.079 | 0.079 | 0.119 |
| **m4_yearly** | 0.118 | 0.114 | 0.123 | 0.122 | 0.121 | 0.129 | 0.132 | 0.113 | **0.091** | 0.125 | **0.111** | 0.161 |
| **m5** | 0.563 | 0.561 | 0.562 | **0.561** | 0.562 | 0.566 | 0.584 | 0.572 | **0.557** | 0.617 | 0.628 | 1.024 |
| **nn5** | 0.190 | 0.190 | 0.188 | 0.190 | **0.150** | 0.157 | 0.157 | **0.147** | 0.155 | 0.248 | 0.264 | 0.425 |
| **nn5_weekly** | 0.086 | 0.084 | 0.085 | 0.085 | **0.084** | 0.085 | 0.090 | 0.090 | **0.079** | 0.084 | 0.088 | 0.123 |
| **tourism_monthly** | 0.100 | 0.100 | 0.096 | 0.095 | **0.090** | 0.098 | 0.095 | 0.104 | **0.085** | 0.091 | 0.100 | 0.104 |
| **tourism_quarterly** | 0.076 | 0.076 | 0.071 | 0.071 | **0.065** | **0.065** | 0.067 | 0.082 | 0.070 | 0.100 | 0.071 | 0.119 |
| **tourism_yearly** | 0.148 | 0.155 | **0.133** | 0.139 | 0.166 | 0.174 | 0.170 | 0.143 | 0.163 | **0.129** | 0.157 | 0.209 |
| **traffic** | 0.257 | 0.254 | 0.255 | 0.250 | **0.231** | 0.244 | 0.253 | 0.234 | **0.212** | 0.354 | 0.558 | 0.362 |
| **weather** | 0.134 | 0.134 | 0.135 | 0.132 | 0.134 | **0.130** | 0.138 | **0.132** | 0.133 | 0.215 | 0.215 | 0.217 |

Table 11: Mean absolute scaled errors (MASE) on Chronos Benchmark II

| | Chroma (4m, freq, best) | Chroma (4m, freq, ens.) | Chroma (tiny, freq, best) | Chroma (tiny, freq, ens.) | Chronos Bolt Base | Chronos Bolt Mini | Chronos Large | Moirai Large | TimesFM 2.0 | Auto ARIMA | Auto ETS | Seasonal Naive |
|---|---|---|---|---|---|---|---|---|---|---|---|---|
| **australian_electricity** | 0.038 | 0.037 | 0.043 | 0.042 | **0.036** | 0.045 | 0.080 | **0.035** | 0.067 | 0.067 | 0.191 | 0.084 |
| **car_parts** | 1.010 | 1.013 | 1.005 | 1.000 | **0.995** | 1.004 | 1.054 | **0.975** | 1.652 | 1.333 | 1.340 | 1.600 |
| **cif_2016** | 0.017 | 0.017 | 0.016 | 0.017 | 0.016 | 0.018 | **0.014** | **0.014** | 0.053 | 0.033 | 0.039 | 0.015 |
| **covid_deaths** | 0.056 | 0.058 | 0.056 | 0.052 | 0.047 | 0.061 | 0.058 | 0.038 | 0.215 | **0.029** | **0.032** | 0.133 |
| **dominick** | 0.361 | 0.362 | 0.360 | 0.363 | 0.345 | 0.350 | **0.330** | **0.332** | 0.371 | 0.485 | 0.484 | 0.453 |
| **ercot** | 0.017 | 0.017 | 0.018 | 0.018 | 0.021 | 0.024 | **0.017** | **0.017** | 0.021 | 0.041 | 0.118 | 0.037 |
| **etth** | **0.075** | **0.075** | 0.079 | 0.079 | **0.071** | 0.077 | 0.076 | 0.090 | 0.085 | 0.089 | 0.132 | 0.122 |
| **ettm** | 0.062 | 0.059 | 0.072 | 0.067 | 0.052 | **0.051** | 0.071 | 0.073 | **0.052** | 0.105 | 0.085 | 0.141 |
| **exchange_rate** | **0.009** | 0.010 | 0.011 | 0.011 | 0.012 | **0.009** | 0.012 | 0.012 | 0.015 | 0.011 | 0.010 | 0.013 |
| **fred_md** | 0.036 | 0.047 | 0.048 | 0.042 | 0.042 | 0.033 | **0.027** | 0.052 | **0.027** | 0.035 | 0.033 | 0.122 |
| **hospital** | 0.054 | 0.054 | 0.054 | 0.054 | 0.057 | 0.058 | 0.055 | **0.053** | **0.050** | 0.059 | 0.053 | 0.073 |
| **m1_monthly** | 0.142 | 0.140 | 0.140 | 0.139 | 0.139 | 0.133 | **0.130** | 0.167 | **0.130** | 0.154 | 0.165 | 0.191 |
| **m1_quarterly** | **0.081** | 0.085 | **0.083** | **0.083** | 0.101 | 0.098 | 0.103 | 0.110 | 0.113 | 0.088 | 0.085 | 0.150 |
| **m1_yearly** | 0.133 | 0.133 | 0.161 | 0.161 | 0.151 | 0.162 | 0.185 | **0.126** | 0.145 | **0.133** | 0.139 | 0.209 |
| **m3_monthly** | 0.094 | 0.094 | **0.093** | **0.093** | 0.093 | 0.095 | 0.095 | 0.097 | **0.089** | 0.098 | 0.093 | 0.149 |
| **m3_quarterly** | 0.078 | 0.075 | 0.072 | 0.072 | 0.076 | 0.078 | 0.073 | **0.070** | 0.075 | 0.077 | **0.070** | 0.101 |
| **m3_yearly** | **0.126** | **0.124** | 0.148 | 0.142 | 0.129 | 0.157 | 0.147 | 0.131 | 0.144 | 0.156 | 0.129 | 0.167 |
| **m4_quarterly** | 0.079 | 0.077 | 0.076 | **0.075** | 0.077 | 0.078 | 0.082 | 0.076 | **0.062** | 0.079 | 0.079 | 0.119 |
| **m4_yearly** | 0.118 | 0.114 | 0.123 | 0.122 | 0.121 | 0.129 | 0.132 | 0.113 | **0.091** | 0.125 | **0.111** | 0.161 |
| **m5** | 0.563 | 0.561 | 0.562 | **0.561** | 0.562 | 0.566 | 0.584 | 0.572 | **0.557** | 0.617 | 0.628 | 1.024 |
| **nn5** | 0.190 | 0.190 | 0.188 | 0.190 | **0.150** | 0.157 | 0.157 | **0.147** | 0.155 | 0.248 | 0.264 | 0.425 |
| **nn5_weekly** | 0.086 | 0.084 | 0.085 | 0.085 | **0.084** | 0.085 | 0.090 | 0.090 | **0.079** | 0.084 | 0.088 | 0.123 |
| **tourism_monthly** | 0.100 | 0.100 | 0.096 | 0.095 | **0.090** | 0.098 | 0.095 | 0.104 | **0.085** | 0.091 | 0.100 | 0.104 |
| **tourism_quarterly** | 0.076 | 0.076 | 0.071 | 0.071 | **0.065** | **0.065** | 0.067 | 0.082 | 0.070 | 0.100 | 0.071 | 0.119 |
| **tourism_yearly** | 0.148 | 0.155 | **0.133** | 0.139 | 0.166 | 0.174 | 0.170 | 0.143 | 0.163 | **0.129** | 0.157 | 0.209 |
| **traffic** | 0.257 | 0.254 | 0.255 | 0.250 | **0.231** | 0.244 | 0.253 | 0.234 | **0.212** | 0.354 | 0.558 | 0.362 |
| **weather** | 0.134 | 0.134 | 0.135 | 0.132 | 0.134 | **0.130** | 0.138 | **0.132** | 0.133 | 0.215 | 0.215 | 0.217 |

Table 12: Weighted quantile losses (WQL) on GIFT-Eval (Part 1)

| | Chroma (4m, freq, best) | Chroma (4m, freq, ens.) | Chroma (tiny, freq, best) | Chroma (tiny, freq, ens.) | Chronos Bolt Base | Chronos Bolt Small | Chronos Large | TTM-R2 Zeroshot | TabPFN-TS | TimesFM 2.0 (500m) | Auto ARIMA | Auto ETS | Seasonal Naïve |
|---|---|---|---|---|---|---|---|---|---|---|---|---|---|
| **LOOP-SEATTLE-5T-48** | 0.064 | 0.062 | 0.077 | 0.065 | 0.055 | 0.055 | 0.070 | 0.068 | 0.052 | **0.051** | 0.081 | 0.083 | 0.081 |
| **LOOP-SEATTLE-5T-480** | 0.165 | 0.158 | 0.146 | 0.138 | 0.116 | 0.119 | 0.176 | 0.121 | **0.087** | 0.110 | 0.123 | 0.238 | 0.123 |
| **LOOP-SEATTLE-5T-720** | 0.139 | 0.131 | 0.133 | 0.125 | 0.129 | 0.125 | 0.143 | 0.125 | **0.094** | 0.114 | 0.137 | 0.240 | 0.137 |
| **LOOP-SEATTLE-D-30** | 0.052 | 0.052 | 0.049 | 0.049 | **0.044** | 0.045 | 0.045 | 0.101 | 0.044 | **0.041** | 0.078 | 0.058 | 0.131 |
| **LOOP-SEATTLE-H-48** | 0.075 | 0.076 | 0.072 | 0.072 | 0.065 | 0.066 | 0.066 | 0.095 | **0.064** | **0.059** | 0.108 | 0.157 | 0.108 |
| **LOOP-SEATTLE-H-480** | 0.086 | 0.084 | 0.082 | 0.083 | 0.076 | 0.082 | 0.084 | 0.104 | **0.065** | **0.067** | 0.154 | 0.339 | 0.206 |
| **LOOP-SEATTLE-H-720** | 0.085 | 0.083 | 0.083 | 0.082 | 0.076 | 0.082 | 0.081 | 0.097 | **0.063** | **0.066** | 0.193 | 0.426 | 0.245 |
| **M-DENSE-D-30** | 0.082 | 0.078 | 0.087 | 0.087 | 0.069 | 0.072 | 0.075 | 0.151 | **0.057** | **0.060** | 0.135 | 0.123 | 0.294 |
| **M-DENSE-H-48** | 0.149 | 0.149 | 0.145 | 0.142 | 0.125 | **0.133** | 0.137 | 0.225 | 0.154 | 0.139 | 0.281 | 0.242 | 0.281 |
| **M-DENSE-H-480** | 0.147 | 0.146 | 0.141 | 0.140 | 0.157 | **0.134** | 0.134 | 0.213 | 0.159 | **0.127** | 0.255 | 0.458 | 0.479 |
| **M-DENSE-H-720** | 0.166 | 0.166 | 0.154 | 0.173 | 0.170 | 0.146 | **0.139** | 0.222 | 0.164 | **0.127** | 0.270 | 0.588 | 0.552 |
| **SZ-TAXI-15T-48** | 0.205 | 0.205 | 0.204 | 0.204 | **0.202** | 0.203 | 0.236 | 0.268 | 0.215 | **0.199** | 0.309 | 0.291 | 0.309 |
| **SZ-TAXI-15T-480** | 0.250 | **0.240** | 0.244 | 0.241 | 0.244 | 0.246 | 0.267 | 0.270 | 0.245 | **0.229** | 0.351 | — | 0.454 |
| **SZ-TAXI-15T-720** | 0.248 | **0.242** | 0.255 | 0.249 | 0.248 | 0.245 | 0.265 | 0.260 | 0.248 | **0.227** | 0.398 | — | 0.554 |
| **SZ-TAXI-H-48** | 0.138 | 0.137 | 0.138 | 0.137 | **0.136** | 0.137 | 0.148 | 0.183 | 0.144 | **0.135** | 0.170 | 1.620 | 0.229 |
| **bitbrains-fast-storage-5T-48** | 0.448 | 0.441 | 0.435 | **0.435** | 0.454 | **0.435** | 0.465 | 0.596 | 0.456 | 0.447 | 1.210 | 1.210 | 1.210 |
| **bitbrains-fast-storage-5T-480** | 0.801 | 0.792 | 0.825 | 0.803 | **0.755** | 0.867 | 0.798 | 0.906 | **0.735** | 0.881 | 1.270 | 1.270 | 1.270 |
| **bitbrains-fast-storage-5T-720** | 0.734 | 0.735 | 0.738 | **0.726** | 0.748 | 0.753 | **0.723** | 0.939 | 0.760 | 0.908 | 1.290 | 1.290 | 1.290 |
| **bitbrains-fast-storage-H-48** | 0.731 | 0.699 | 0.686 | 0.671 | 0.774 | **0.589** | 0.633 | 0.926 | **0.591** | 0.688 | 0.844 | — | 1.080 |
| **bitbrains-rnd-5T-48** | 0.441 | 0.441 | **0.430** | **0.435** | 0.438 | 0.453 | 0.506 | 0.605 | 0.455 | 0.461 | 1.100 | 1.100 | 1.100 |
| **bitbrains-rnd-5T-480** | 0.751 | 0.729 | 0.743 | 0.733 | **0.605** | 0.792 | **0.636** | 0.835 | 0.710 | 0.727 | 1.260 | 1.260 | 1.260 |
| **bitbrains-rnd-5T-720** | **0.704** | 0.730 | 0.862 | 0.810 | 0.756 | 0.756 | 1.070 | 0.779 | 0.730 | **0.706** | 1.290 | 1.290 | 1.290 |
| **bitbrains-rnd-H-48** | 0.796 | 0.764 | 0.706 | 0.685 | 0.624 | **0.623** | 0.661 | 1.060 | 0.637 | 0.649 | 0.874 | 1.300 | 1.300 |
| **bizitobs-application-60** | 0.037 | 0.026 | 0.031 | **0.021** | 0.054 | 0.035 | 0.032 | 0.058 | 0.031 | **0.014** | 0.035 | 0.048 | 0.035 |
| **bizitobs-application-600** | 0.106 | 0.095 | 0.085 | 0.082 | 0.104 | 0.085 | 0.095 | 0.126 | 0.070 | **0.033** | **0.042** | 0.161 | **0.042** |
| **bizitobs-application-900** | 0.119 | 0.110 | 0.094 | 0.093 | 0.109 | 0.092 | **0.084** | 0.144 | 0.088 | **0.057** | 0.973 | 0.269 | 0.973 |
| **bizitobs-12c-5T-48** | 0.079 | 0.079 | 0.078 | 0.077 | 0.074 | **0.073** | 0.084 | 0.107 | 0.099 | 0.084 | 0.262 | **0.075** | 0.262 |
| **bizitobs-12c-5T-480** | 0.467 | 0.449 | 0.426 | 0.417 | 0.445 | 0.462 | 0.496 | 0.540 | **0.345** | 0.529 | 0.530 | **0.355** | 0.530 |
| **bizitobs-12c-5T-720** | 0.785 | 0.737 | 0.673 | 0.649 | 0.738 | 0.790 | 0.741 | 0.785 | **0.511** | 0.748 | 0.674 | **0.597** | 0.674 |
| **bizitobs-12c-H-48** | 0.221 | 0.215 | 0.209 | 0.210 | **0.189** | **0.204** | 0.491 | 0.549 | 0.290 | 0.345 | 0.547 | 0.494 | 0.536 |
| **bizitobs-12c-H-480** | 0.288 | 0.290 | **0.277** | 0.278 | **0.254** | 0.285 | 0.802 | 0.782 | 0.380 | 0.640 | 0.813 | 1.050 | 1.420 |
| **bizitobs-12c-H-720** | 0.314 | 0.310 | 0.308 | 0.300 | **0.278** | 0.295 | 0.757 | 0.751 | 0.440 | 0.728 | 0.787 | 1.110 | 1.820 |
| **bizitobs-service-60** | 0.032 | 0.023 | 0.025 | **0.019** | 0.051 | 0.032 | 0.027 | 0.053 | 0.031 | **0.015** | 0.040 | 0.046 | 0.040 |
| **bizitobs-service-600** | 0.091 | 0.081 | 0.074 | 0.075 | 0.096 | 0.082 | 0.072 | 0.117 | 0.067 | **0.038** | **0.049** | 0.224 | **0.049** |
| **bizitobs-service-900** | 0.115 | 0.106 | 0.097 | 0.095 | 0.113 | 0.096 | 0.094 | 0.140 | 0.091 | 0.062 | **0.056** | 0.373 | **0.056** |
| **car-parts-with-missing-12** | 1.010 | 1.013 | 1.005 | 1.000 | 0.995 | 1.007 | 1.070 | 2.285 | **0.955** | 1.046 | 1.290 | 1.340 | 1.720 |
| **covid-deaths-30** | 0.056 | 0.058 | 0.056 | 0.052 | 0.047 | 0.043 | 0.042 | 0.123 | 0.040 | 0.062 | **0.030** | **0.033** | 0.125 |
| **electricity-15T-48** | 0.093 | 0.092 | 0.091 | 0.090 | 0.082 | 0.082 | 0.092 | 0.152 | 0.104 | **0.079** | 0.165 | 0.175 | 0.165 |
| **electricity-15T-480** | 0.092 | 0.088 | 0.088 | 0.086 | 0.083 | 0.087 | 0.094 | 0.142 | 0.092 | **0.080** | 0.124 | 0.124 | 0.124 |
| **electricity-15T-720** | 0.092 | 0.089 | 0.092 | 0.087 | **0.084** | 0.086 | 0.094 | 0.143 | 0.089 | 0.083 | 0.129 | 0.129 | 0.129 |
| **electricity-D-30** | 0.058 | 0.058 | 0.056 | **0.056** | **0.055** | 0.058 | 0.061 | 0.093 | 0.060 | 0.060 | 0.083 | 0.111 | 0.122 |
| **electricity-H-48** | 0.075 | 0.074 | 0.073 | 0.073 | **0.064** | 0.067 | 0.065 | 0.097 | 0.072 | **0.054** | 0.109 | 0.099 | 0.109 |
| **electricity-H-480** | 0.095 | 0.094 | 0.091 | 0.088 | **0.081** | 0.084 | 0.086 | 0.109 | 0.091 | **0.073** | 0.156 | 0.156 | 0.156 |
| **electricity-H-720** | 0.110 | 0.108 | 0.102 | 0.109 | **0.098** | 0.102 | 0.107 | 0.128 | 0.112 | **0.089** | 0.190 | — | 0.190 |
| **electricity-W-8** | 0.120 | 0.089 | 0.114 | 0.086 | **0.047** | **0.048** | 0.049 | 0.159 | 0.051 | 0.049 | 0.100 | 0.098 | 0.099 |
| **ett1-15T-48** | 0.173 | 0.170 | 0.171 | 0.169 | **0.158** | 0.169 | 0.196 | 0.235 | 0.183 | **0.168** | 0.241 | 0.383 | 0.241 |
| **ett1-15T-480** | 0.290 | 0.282 | 0.306 | 0.276 | 0.281 | 0.288 | 0.376 | 0.333 | **0.248** | 0.278 | 0.352 | 0.867 | 0.352 |
| **ett1-15T-720** | 0.289 | 0.284 | 0.295 | 0.295 | 0.298 | 0.296 | 0.367 | 0.352 | **0.260** | **0.283** | 0.396 | 1.050 | 0.396 |
| **ett1-D-30** | 0.311 | 0.311 | 0.275 | **0.273** | 0.287 | 0.283 | 0.404 | 0.416 | 0.292 | 0.281 | **0.279** | 0.334 | 0.515 |
| **ett1-H-48** | 0.197 | 0.195 | 0.193 | 0.192 | **0.181** | **0.189** | 0.198 | 0.250 | 0.194 | 0.192 | 0.223 | 0.440 | 0.250 |

Table 13: Weighted quantile losses (WQL) on GIFT-Eval (Part 2)

| | Chroma (4m, freq, best) | Chroma (4m, freq, ens.) | Chroma (tiny, freq, best) | Chroma (tiny, freq, ens.) | Chronos Bolt Base | Chronos Bolt Small | Chronos Large | TTM-R2 Zeroshot | TabPFN-TS | TimesFM 2.0 (500m) | Auto ARIMA | Auto ETS | Seasonal Naive |
|---|---|---|---|---|---|---|---|---|---|---|---|---|---|
| ett1-H-480 | 0.309 | 0.318 | 0.298 | 0.343 | 0.303 | 0.295 | 0.333 | 0.339 | **0.276** | **0.282** | 0.384 | — | 0.540 |
| ett1-H-720 | 0.592 | 0.541 | **0.305** | 0.467 | 0.311 | 0.337 | 0.350 | 0.342 | **0.290** | 0.310 | 0.430 | — | 0.616 |
| ett1-W-8 | 0.281 | **0.253** | 0.277 | 0.260 | 0.296 | 0.293 | 0.316 | 0.448 | 0.256 | 0.272 | 0.305 | 0.277 | 0.338 |
| ett2-15T-48 | 0.071 | 0.068 | **0.064** | **0.064** | 0.067 | 0.070 | 0.072 | 0.093 | 0.073 | 0.065 | 0.096 | 0.103 | 0.096 |
| ett2-15T-480 | 0.119 | 0.115 | 0.109 | 0.107 | 0.110 | 0.119 | 0.126 | 0.128 | **0.098** | **0.105** | 0.143 | 0.325 | 0.143 |
| ett2-15T-720 | 0.121 | 0.117 | 0.117 | 0.113 | 0.111 | 0.118 | 0.140 | 0.126 | **0.101** | **0.106** | 0.165 | 0.317 | 0.165 |
| ett2-D-30 | 0.096 | 0.099 | 0.096 | 0.104 | 0.094 | **0.091** | **0.091** | 0.119 | 0.129 | 0.108 | 0.125 | 0.158 | 0.205 |
| ett2-H-48 | 0.066 | 0.066 | 0.067 | 0.067 | **0.063** | 0.065 | 0.072 | 0.088 | 0.070 | 0.066 | 0.089 | 0.103 | 0.094 |
| ett2-H-480 | 0.129 | 0.118 | 0.121 | 0.121 | **0.115** | 0.118 | 0.125 | 0.139 | 0.128 | **0.110** | 0.245 | 0.272 | 0.241 |
| ett2-H-720 | 0.169 | 0.146 | 0.129 | 0.130 | **0.117** | **0.121** | 0.139 | 0.144 | 0.136 | 0.125 | 0.272 | 0.329 | 0.287 |
| ett2-W-8 | 0.093 | 0.094 | 0.099 | 0.095 | 0.088 | 0.094 | **0.071** | 0.200 | 0.120 | 0.110 | 0.136 | 0.115 | 0.169 |
| hierarchical-sales-D-30 | 0.577 | 0.576 | 0.580 | 0.577 | **0.576** | 0.582 | 0.599 | 0.792 | 0.593 | **0.576** | 0.735 | 0.931 | 2.360 |
| hierarchical-sales-W-8 | 0.350 | 0.350 | 0.356 | 0.351 | 0.353 | 0.354 | 0.365 | 0.725 | **0.342** | **0.330** | 0.485 | 0.642 | 1.030 |
| hospital-12 | 0.054 | 0.054 | 0.054 | 0.054 | 0.057 | 0.058 | 0.057 | 0.123 | **0.050** | **0.050** | 0.060 | 0.053 | 0.062 |
| jena-weather-10T-48 | 0.035 | 0.034 | 0.046 | 0.048 | 0.033 | 0.037 | 0.044 | 0.045 | 0.035 | **0.016** | 0.155 | 0.082 | 0.155 |
| jena-weather-10T-480 | 0.059 | 0.060 | 0.058 | 0.057 | 0.057 | 0.060 | 0.075 | 0.069 | 0.057 | **0.031** | 0.277 | 0.210 | 0.277 |
| jena-weather-10T-720 | 0.060 | 0.059 | 0.070 | 0.067 | 0.064 | 0.063 | 0.077 | 0.068 | **0.055** | **0.035** | 0.304 | 0.282 | 0.304 |
| jena-weather-D-30 | 0.046 | 0.046 | 0.046 | 0.046 | 0.045 | 0.047 | 0.049 | 0.124 | 0.047 | 0.058 | 0.080 | 0.074 | 0.297 |
| jena-weather-H-48 | 0.043 | 0.046 | **0.042** | 0.044 | **0.042** | 0.043 | 0.046 | 0.060 | 0.043 | 0.045 | 0.143 | 0.158 | 0.173 |
| jena-weather-H-480 | **0.055** | **0.055** | 0.059 | 0.059 | **0.054** | 0.058 | 0.071 | 0.073 | 0.060 | 0.066 | 0.211 | — | 0.486 |
| jena-weather-H-720 | **0.066** | 0.074 | 0.088 | 0.083 | **0.062** | 0.068 | 0.072 | 0.084 | 0.066 | 0.068 | 0.230 | — | 0.598 |
| kdd-cup-2018-with-missing-D-30 | 0.375 | **0.368** | 0.379 | 0.375 | 0.372 | 0.373 | 0.502 | 0.452 | **0.359** | 0.378 | 0.393 | — | 0.888 |
| kdd-cup-2018-with-missing-H-48 | 0.441 | 0.398 | 0.357 | 0.380 | **0.246** | 0.267 | 0.458 | 0.514 | 0.437 | 0.376 | 0.559 | 1.160 | 0.559 |
| kdd-cup-2018-with-missing-H-480 | 0.456 | 0.443 | 0.423 | 0.425 | **0.301** | 0.364 | 0.658 | 0.532 | 0.462 | 0.466 | 0.851 | 0.949 | 0.949 |
| kdd-cup-2018-with-missing-H-720 | 0.494 | 0.498 | 0.471 | 0.471 | **0.300** | 0.419 | 0.636 | 0.542 | 0.462 | 0.518 | 1.050 | 1.250 | 1.250 |
| m4-daily-14 | 0.024 | 0.023 | 0.023 | 0.022 | 0.021 | 0.021 | 0.022 | 0.035 | 0.024 | **0.021** | 0.023 | 0.029 | 0.026 |
| m4-hourly-48 | 0.028 | 0.029 | 0.027 | 0.026 | 0.025 | **0.020** | 0.026 | 0.040 | 0.028 | **0.011** | 0.034 | 0.070 | 0.040 |
| m4-monthly-18 | 0.095 | 0.094 | 0.094 | 0.093 | 0.094 | 0.094 | 0.103 | 0.177 | **0.088** | **0.067** | 0.098 | 0.100 | 0.126 |
| m4-quarterly-8 | 0.079 | 0.077 | 0.076 | 0.075 | 0.077 | 0.078 | 0.083 | 0.139 | 0.075 | **0.062** | 0.082 | 0.079 | 0.099 |
| m4-weekly-13 | 0.043 | 0.044 | 0.043 | 0.043 | 0.038 | 0.038 | **0.037** | 0.069 | **0.036** | 0.042 | 0.050 | 0.052 | 0.073 |
| m4-yearly-6 | 0.118 | 0.114 | 0.123 | 0.122 | 0.121 | 0.128 | 0.135 | 0.197 | 0.113 | **0.091** | 0.130 | **0.111** | 0.138 |
| restaurant-30 | 0.287 | 0.282 | 0.279 | 0.278 | **0.264** | **0.264** | 0.279 | 0.438 | 0.297 | **0.261** | 0.362 | — | 0.907 |
| saugeenday-D-30 | 0.440 | 0.437 | 0.388 | 0.387 | **0.338** | 0.354 | 0.420 | 0.589 | 0.384 | 0.408 | 0.564 | 0.596 | 0.754 |
| saugeenday-M-12 | 0.334 | 0.334 | 0.324 | 0.379 | 0.296 | **0.293** | 0.415 | 0.405 | **0.278** | 0.342 | 0.326 | 0.322 | 0.445 |
| saugeenday-W-8 | 0.506 | 0.479 | 0.449 | 0.449 | **0.363** | **0.372** | 0.471 | 0.696 | **0.380** | 0.601 | 0.549 | 0.896 | 0.855 |
| solar-10T-48 | 0.658 | 0.532 | **0.475** | 0.504 | 0.511 | **0.498** | 0.575 | 0.785 | 0.545 | 0.804 | 0.860 | 0.870 | 0.860 |
| solar-10T-480 | **0.353** | **0.353** | 0.356 | **0.350** | 0.436 | 0.453 | 0.681 | 0.573 | 0.359 | 0.516 | 0.771 | — | 0.771 |
| solar-10T-720 | 0.355 | 0.355 | **0.340** | **0.334** | 0.443 | 0.497 | 0.747 | 0.545 | 0.352 | 0.498 | 0.786 | 2.930 | 0.786 |
| solar-D-30 | 0.284 | 0.279 | 0.281 | **0.278** | 0.287 | 0.286 | 0.328 | 0.396 | **0.269** | 0.278 | 0.282 | 0.281 | 0.757 |
| solar-H-48 | 0.322 | 0.322 | 0.317 | 0.316 | **0.298** | 0.303 | 0.337 | 0.468 | 0.358 | 0.406 | 0.628 | 1.080 | 0.628 |
| solar-H-480 | 0.336 | 0.333 | 0.332 | **0.331** | 0.368 | 0.356 | 0.355 | 0.493 | **0.324** | 0.376 | 0.557 | 2.120 | 1.270 |
| solar-H-720 | 0.341 | 0.339 | **0.338** | 0.343 | 0.405 | 0.373 | 0.448 | 0.512 | **0.324** | 0.493 | 0.607 | 1.720 | 1.470 |
| solar-W-8 | 0.227 | 0.183 | 0.136 | 0.147 | **0.133** | 0.136 | 0.162 | 0.531 | **0.124** | 0.171 | 0.152 | 0.139 | 0.236 |
| temperature-rain-with-missing-30 | 0.551 | 0.551 | 0.548 | 0.548 | **0.538** | 0.544 | 0.609 | 0.791 | 0.565 | 0.586 | 0.694 | 0.754 | 1.630 |
| us-births-D-30 | 0.023 | 0.023 | 0.022 | 0.023 | 0.026 | 0.028 | 0.023 | 0.104 | **0.018** | **0.019** | 0.074 | 0.074 | 0.144 |
| us-births-M-12 | 0.021 | 0.021 | 0.019 | 0.020 | 0.019 | 0.016 | 0.013 | 0.036 | 0.013 | **0.011** | **0.010** | 0.012 | 0.017 |
| us-births-W-8 | 0.016 | 0.016 | 0.015 | 0.015 | 0.013 | 0.013 | **0.010** | 0.027 | **0.011** | 0.013 | 0.018 | 0.018 | 0.022 |

Table 14: Mean absolute scaled errors (MASE) on GIFT-Eval (Part 1)

| | Chroma (4m, freq, best) | Chroma (4m, freq, ens.) | Chroma (tiny, freq, best) | Chroma (tiny, freq, ens.) | Chronos Bolt Base | Chronos Bolt Small | Chronos Large | TTM-R2 Zeroshot | TabPFN-TS | TimesFM 2.0 (500m) | Auto ARIMA | Auto ETS | Seasonal Naïve |
|---|---|---|---|---|---|---|---|---|---|---|---|---|---|
| **LOOP-SEATTLE-5T-48** | 0.064 | 0.062 | 0.077 | 0.065 | 0.055 | 0.055 | 0.070 | 0.068 | **0.052** | **0.051** | 0.081 | 0.083 | 0.081 |
| **LOOP-SEATTLE-5T-480** | 0.165 | 0.158 | 0.146 | 0.138 | 0.116 | 0.119 | 0.176 | 0.121 | **0.087** | 0.110 | 0.123 | 0.238 | 0.123 |
| **LOOP-SEATTLE-5T-720** | 0.139 | 0.131 | 0.133 | 0.125 | 0.129 | 0.125 | 0.143 | 0.125 | **0.094** | 0.114 | 0.137 | 0.240 | 0.137 |
| **LOOP-SEATTLE-D-30** | 0.052 | 0.052 | 0.049 | 0.049 | **0.044** | 0.045 | 0.045 | 0.101 | 0.044 | **0.041** | 0.078 | 0.058 | 0.131 |
| **LOOP-SEATTLE-H-48** | 0.075 | 0.076 | 0.072 | 0.072 | 0.065 | 0.066 | 0.066 | 0.095 | **0.064** | **0.059** | 0.108 | 0.157 | 0.108 |
| **LOOP-SEATTLE-H-480** | 0.086 | 0.084 | 0.082 | 0.083 | 0.076 | 0.082 | 0.084 | 0.104 | **0.065** | **0.067** | 0.154 | 0.339 | 0.206 |
| **LOOP-SEATTLE-H-720** | 0.085 | 0.083 | 0.083 | 0.082 | 0.076 | 0.082 | 0.081 | 0.097 | **0.063** | **0.066** | 0.193 | 0.426 | 0.245 |
| **M-DENSE-D-30** | 0.082 | 0.078 | 0.087 | 0.087 | 0.069 | 0.072 | 0.075 | 0.151 | **0.057** | **0.060** | 0.135 | 0.123 | 0.294 |
| **M-DENSE-H-48** | 0.149 | 0.149 | 0.145 | 0.142 | 0.125 | **0.133** | 0.137 | 0.225 | 0.154 | 0.139 | 0.281 | 0.242 | 0.281 |
| **M-DENSE-H-480** | 0.147 | 0.146 | 0.141 | 0.140 | 0.157 | **0.134** | 0.134 | 0.213 | 0.159 | **0.127** | 0.255 | 0.458 | 0.479 |
| **M-DENSE-H-720** | 0.166 | 0.166 | 0.154 | 0.173 | 0.170 | 0.146 | **0.139** | 0.222 | 0.164 | **0.127** | 0.270 | 0.588 | 0.552 |
| **SZ-TAXI-15T-48** | 0.205 | 0.205 | 0.204 | 0.204 | **0.202** | 0.203 | 0.236 | 0.268 | 0.215 | **0.199** | 0.309 | 0.291 | 0.309 |
| **SZ-TAXI-15T-480** | 0.250 | **0.240** | 0.244 | 0.241 | 0.244 | 0.246 | 0.267 | 0.270 | 0.245 | **0.229** | 0.351 | — | 0.454 |
| **SZ-TAXI-15T-720** | 0.248 | **0.242** | 0.255 | 0.249 | 0.248 | 0.245 | 0.265 | 0.260 | 0.248 | **0.227** | 0.398 | — | 0.554 |
| **SZ-TAXI-H-48** | 0.138 | 0.137 | 0.138 | 0.137 | **0.136** | 0.137 | 0.148 | 0.183 | 0.144 | **0.135** | 0.170 | 1.620 | 0.229 |
| **bitbrains-fast-storage-5T-48** | 0.448 | 0.441 | **0.435** | **0.435** | 0.454 | **0.435** | 0.465 | 0.596 | 0.456 | 0.447 | 1.210 | 1.210 | 1.210 |
| **bitbrains-fast-storage-5T-480** | 0.801 | 0.792 | 0.825 | 0.803 | **0.755** | 0.867 | 0.798 | 0.906 | **0.735** | 0.881 | 1.270 | 1.270 | 1.270 |
| **bitbrains-fast-storage-5T-720** | 0.734 | 0.735 | 0.738 | **0.726** | 0.748 | 0.753 | **0.723** | 0.939 | 0.760 | 0.908 | 1.290 | 1.290 | 1.290 |
| **bitbrains-fast-storage-H-48** | 0.731 | 0.699 | 0.686 | 0.671 | 0.774 | **0.589** | 0.633 | 0.926 | **0.591** | 0.688 | 0.844 | — | 1.080 |
| **bitbrains-rnd-5T-48** | 0.441 | 0.441 | **0.430** | **0.435** | 0.438 | 0.453 | 0.506 | 0.605 | 0.455 | 0.461 | 1.100 | 1.100 | 1.100 |
| **bitbrains-rnd-5T-480** | 0.751 | 0.729 | 0.743 | 0.733 | **0.605** | 0.792 | **0.636** | 0.835 | 0.710 | 0.727 | 1.260 | 1.260 | 1.260 |
| **bitbrains-rnd-5T-720** | **0.704** | 0.730 | 0.862 | 0.810 | 0.756 | 0.756 | 1.070 | 0.779 | 0.730 | **0.706** | 1.290 | 1.290 | 1.290 |
| **bitbrains-rnd-H-48** | 0.796 | 0.764 | 0.706 | 0.685 | 0.624 | **0.623** | 0.661 | 1.060 | 0.637 | 0.649 | 0.874 | 1.300 | 1.300 |
| **bizitobs-application-60** | 0.037 | 0.026 | 0.031 | **0.021** | 0.054 | 0.035 | 0.032 | 0.058 | 0.031 | **0.014** | 0.035 | 0.048 | 0.035 |
| **bizitobs-application-600** | 0.106 | 0.095 | 0.085 | 0.082 | 0.104 | 0.085 | 0.095 | 0.126 | 0.070 | **0.033** | **0.042** | 0.161 | **0.042** |
| **bizitobs-application-900** | 0.119 | 0.110 | 0.094 | 0.093 | 0.109 | 0.092 | **0.084** | 0.144 | 0.088 | **0.057** | 0.973 | 0.269 | 0.973 |
| **bizitobs-l2c-5T-48** | 0.079 | 0.079 | 0.078 | 0.077 | 0.074 | **0.073** | 0.084 | 0.107 | 0.099 | 0.084 | 0.262 | **0.075** | 0.262 |
| **bizitobs-l2c-5T-480** | 0.467 | 0.449 | 0.426 | 0.417 | 0.445 | 0.462 | 0.496 | 0.540 | **0.345** | 0.529 | 0.530 | **0.355** | 0.530 |
| **bizitobs-l2c-5T-720** | 0.785 | 0.737 | 0.673 | 0.649 | 0.738 | 0.790 | 0.741 | 0.785 | **0.511** | 0.748 | 0.674 | **0.597** | 0.674 |
| **bizitobs-l2c-H-48** | 0.221 | 0.215 | 0.209 | 0.210 | **0.189** | **0.204** | 0.491 | 0.549 | 0.290 | 0.345 | 0.547 | 0.494 | 0.536 |
| **bizitobs-l2c-H-480** | 0.288 | 0.290 | **0.277** | 0.278 | **0.254** | 0.285 | 0.802 | 0.782 | 0.380 | 0.640 | 0.813 | 1.050 | 1.420 |
| **bizitobs-l2c-H-720** | 0.314 | 0.310 | 0.308 | 0.300 | **0.278** | 0.295 | 0.757 | 0.751 | 0.440 | 0.728 | 0.787 | 1.110 | 1.820 |
| **bizitobs-service-60** | 0.032 | 0.023 | 0.025 | **0.019** | 0.051 | 0.032 | 0.027 | 0.053 | 0.031 | **0.015** | 0.040 | 0.046 | 0.040 |
| **bizitobs-service-600** | 0.091 | 0.081 | 0.074 | 0.075 | 0.096 | 0.082 | 0.072 | 0.117 | 0.067 | **0.038** | **0.049** | 0.224 | **0.049** |
| **bizitobs-service-900** | 0.115 | 0.106 | 0.097 | 0.095 | 0.113 | 0.096 | 0.094 | 0.140 | 0.091 | 0.062 | **0.056** | 0.373 | **0.056** |
| **car-parts-with-missing-12** | 1.010 | 1.013 | 1.005 | 1.000 | 0.995 | 1.007 | 1.070 | 2.285 | **0.955** | 1.046 | 1.290 | 1.340 | 1.720 |
| **covid-deaths-30** | 0.056 | 0.058 | 0.056 | 0.052 | 0.047 | 0.043 | 0.042 | 0.123 | 0.040 | 0.062 | **0.030** | **0.033** | 0.125 |
| **electricity-15T-48** | 0.093 | 0.092 | 0.091 | 0.090 | 0.082 | 0.082 | 0.092 | 0.152 | 0.104 | **0.079** | 0.165 | 0.175 | 0.165 |
| **electricity-15T-480** | 0.092 | 0.088 | 0.088 | 0.086 | 0.083 | 0.087 | 0.094 | 0.142 | 0.092 | **0.080** | 0.124 | 0.124 | 0.124 |
| **electricity-15T-720** | 0.092 | 0.089 | 0.092 | 0.087 | **0.084** | 0.086 | 0.094 | 0.143 | 0.089 | **0.083** | 0.129 | 0.129 | 0.129 |
| **electricity-D-30** | 0.058 | 0.058 | 0.056 | **0.056** | **0.055** | 0.058 | 0.061 | 0.093 | 0.060 | 0.060 | 0.083 | 0.111 | 0.122 |
| **electricity-H-48** | 0.075 | 0.074 | 0.073 | 0.073 | **0.064** | 0.067 | 0.065 | 0.097 | 0.072 | **0.054** | 0.109 | 0.099 | 0.109 |
| **electricity-H-480** | 0.095 | 0.094 | 0.091 | 0.088 | **0.081** | 0.084 | 0.086 | 0.109 | 0.091 | **0.073** | 0.156 | 0.156 | 0.156 |
| **electricity-H-720** | 0.110 | 0.108 | 0.102 | 0.109 | **0.098** | 0.102 | 0.107 | 0.128 | 0.112 | **0.089** | 0.190 | — | 0.190 |
| **electricity-W-8** | 0.120 | 0.089 | 0.114 | 0.086 | **0.047** | **0.048** | 0.049 | 0.159 | 0.051 | 0.049 | 0.100 | 0.098 | 0.099 |
| **ett1-15T-48** | 0.173 | 0.170 | 0.171 | 0.169 | **0.158** | 0.169 | 0.196 | 0.235 | 0.183 | **0.168** | 0.241 | 0.383 | 0.241 |
| **ett1-15T-480** | 0.290 | 0.282 | 0.306 | 0.276 | 0.281 | 0.288 | 0.376 | 0.333 | **0.248** | 0.278 | 0.352 | 0.867 | 0.352 |
| **ett1-15T-720** | 0.289 | 0.284 | 0.295 | 0.295 | 0.298 | 0.296 | 0.367 | 0.352 | **0.260** | 0.283 | 0.396 | 1.050 | 0.396 |
| **ett1-D-30** | 0.311 | 0.311 | 0.295 | **0.273** | 0.287 | 0.283 | 0.404 | 0.416 | 0.292 | 0.281 | **0.279** | 0.334 | 0.515 |
| **ett1-H-48** | 0.197 | 0.195 | 0.193 | 0.192 | **0.181** | **0.189** | 0.198 | 0.250 | 0.194 | 0.192 | 0.223 | 0.440 | 0.250 |

Table 15: Mean absolute scaled errors (MASE) on GIFT-Eval (Part 2)

| | Chroma (4m, freq, best) | Chroma (4m, freq, ens.) | Chroma (tiny, freq, best) | Chroma (tiny, freq, ens.) | Chronos Bolt Base | Chronos Bolt Small | Chronos Large | TTM-R2 Zeroshot | TabPFN-TS | TimesFM 2.0 (500m) | Auto ARIMA | Auto ETS | Seasonal Naïve |
|---|---|---|---|---|---|---|---|---|---|---|---|---|---|
| ett1-H-480 | 0.309 | 0.318 | 0.298 | 0.343 | 0.303 | 0.295 | 0.333 | 0.339 | 0.276 | 0.282 | 0.384 | — | 0.540 |
| ett1-H-720 | 0.592 | 0.541 | 0.305 | 0.467 | 0.311 | 0.337 | 0.350 | 0.342 | 0.290 | 0.310 | 0.430 | — | 0.616 |
| ett1-W-8 | 0.281 | 0.253 | 0.277 | 0.260 | 0.296 | 0.293 | 0.316 | 0.448 | 0.256 | 0.272 | 0.305 | 0.277 | 0.338 |
| ett2-15T-48 | 0.071 | 0.068 | 0.064 | 0.064 | 0.067 | 0.070 | 0.072 | 0.093 | 0.073 | 0.065 | 0.096 | 0.103 | 0.096 |
| ett2-15T-480 | 0.119 | 0.115 | 0.109 | 0.107 | 0.110 | 0.119 | 0.126 | 0.128 | 0.098 | 0.105 | 0.143 | 0.325 | 0.143 |
| ett2-15T-720 | 0.121 | 0.117 | 0.117 | 0.113 | 0.111 | 0.118 | 0.140 | 0.126 | 0.101 | 0.106 | 0.165 | 0.317 | 0.165 |
| ett2-D-30 | 0.096 | 0.099 | 0.096 | 0.104 | 0.094 | 0.091 | 0.091 | 0.119 | 0.129 | 0.108 | 0.125 | 0.158 | 0.205 |
| ett2-H-48 | 0.066 | 0.066 | 0.067 | 0.067 | 0.063 | 0.065 | 0.072 | 0.088 | 0.070 | 0.066 | 0.089 | 0.103 | 0.094 |
| ett2-H-480 | 0.129 | 0.118 | 0.121 | 0.121 | 0.115 | 0.118 | 0.125 | 0.139 | 0.128 | 0.110 | 0.245 | 0.272 | 0.241 |
| ett2-H-720 | 0.169 | 0.146 | 0.129 | 0.130 | 0.117 | 0.121 | 0.139 | 0.144 | 0.136 | 0.125 | 0.272 | 0.329 | 0.287 |
| ett2-W-8 | 0.093 | 0.094 | 0.099 | 0.095 | 0.088 | 0.094 | 0.071 | 0.200 | 0.120 | 0.110 | 0.136 | 0.115 | 0.169 |
| hierarchical-sales-D-30 | 0.577 | 0.576 | 0.580 | 0.577 | 0.576 | 0.582 | 0.599 | 0.792 | 0.593 | 0.576 | 0.735 | 0.931 | 2.360 |
| hierarchical-sales-W-8 | 0.350 | 0.350 | 0.356 | 0.351 | 0.353 | 0.354 | 0.365 | 0.725 | 0.342 | 0.330 | 0.485 | 0.642 | 1.030 |
| hospital-12 | 0.054 | 0.054 | 0.054 | 0.054 | 0.057 | 0.058 | 0.057 | 0.123 | 0.050 | 0.050 | 0.060 | 0.053 | 0.062 |
| jena-weather-10T-48 | 0.035 | 0.034 | 0.046 | 0.048 | 0.033 | 0.037 | 0.044 | 0.045 | 0.035 | 0.016 | 0.155 | 0.082 | 0.155 |
| jena-weather-10T-480 | 0.059 | 0.060 | 0.058 | 0.057 | 0.057 | 0.060 | 0.075 | 0.069 | 0.057 | 0.031 | 0.277 | 0.210 | 0.277 |
| jena-weather-10T-720 | 0.060 | 0.059 | 0.070 | 0.067 | 0.064 | 0.063 | 0.077 | 0.068 | 0.055 | 0.035 | 0.304 | 0.282 | 0.304 |
| jena-weather-D-30 | 0.046 | 0.046 | 0.046 | 0.046 | 0.045 | 0.047 | 0.049 | 0.124 | 0.047 | 0.058 | 0.080 | 0.074 | 0.297 |
| jena-weather-H-48 | 0.043 | 0.046 | 0.042 | 0.044 | 0.042 | 0.043 | 0.046 | 0.060 | 0.043 | 0.045 | 0.143 | 0.158 | 0.173 |
| jena-weather-H-480 | 0.055 | 0.055 | 0.059 | 0.059 | 0.054 | 0.058 | 0.071 | 0.073 | 0.060 | 0.066 | 0.211 | — | 0.486 |
| jena-weather-H-720 | 0.066 | 0.074 | 0.088 | 0.083 | 0.062 | 0.068 | 0.072 | 0.084 | 0.066 | 0.068 | 0.230 | — | 0.598 |
| kdd-cup-2018-with-missing-D-30 | 0.375 | 0.368 | 0.379 | 0.375 | 0.372 | 0.373 | 0.502 | 0.452 | 0.359 | 0.378 | 0.393 | — | 0.888 |
| kdd-cup-2018-with-missing-H-48 | 0.441 | 0.398 | 0.357 | 0.380 | 0.246 | 0.267 | 0.458 | 0.514 | 0.437 | 0.376 | 0.559 | 1.160 | 0.559 |
| kdd-cup-2018-with-missing-H-480 | 0.456 | 0.443 | 0.423 | 0.425 | 0.301 | 0.364 | 0.658 | 0.532 | 0.462 | 0.466 | 0.851 | 0.949 | 0.949 |
| kdd-cup-2018-with-missing-H-720 | 0.494 | 0.498 | 0.471 | 0.471 | 0.300 | 0.419 | 0.636 | 0.542 | 0.462 | 0.518 | 1.050 | 1.250 | 1.250 |
| m4-daily-14 | 0.024 | 0.023 | 0.023 | 0.022 | 0.021 | 0.021 | 0.022 | 0.035 | 0.024 | 0.021 | 0.023 | 0.029 | 0.026 |
| m4-hourly-48 | 0.028 | 0.029 | 0.027 | 0.026 | 0.025 | 0.020 | 0.026 | 0.040 | 0.028 | 0.011 | 0.034 | 0.070 | 0.040 |
| m4-monthly-18 | 0.095 | 0.094 | 0.094 | 0.093 | 0.094 | 0.094 | 0.103 | 0.177 | 0.088 | 0.067 | 0.098 | 0.100 | 0.126 |
| m4-quarterly-8 | 0.079 | 0.077 | 0.076 | 0.075 | 0.077 | 0.078 | 0.083 | 0.139 | 0.075 | 0.062 | 0.082 | 0.079 | 0.099 |
| m4-weekly-13 | 0.043 | 0.044 | 0.043 | 0.043 | 0.038 | 0.038 | 0.037 | 0.069 | 0.036 | 0.042 | 0.050 | 0.052 | 0.073 |
| m4-yearly-6 | 0.118 | 0.114 | 0.123 | 0.122 | 0.121 | 0.128 | 0.135 | 0.197 | 0.113 | 0.091 | 0.130 | 0.111 | 0.138 |
| restaurant-30 | 0.287 | 0.282 | 0.279 | 0.278 | 0.264 | 0.264 | 0.279 | 0.438 | 0.297 | 0.261 | 0.362 | — | 0.907 |
| saugeenday-D-30 | 0.440 | 0.437 | 0.388 | 0.387 | 0.338 | 0.354 | 0.420 | 0.589 | 0.384 | 0.408 | 0.564 | 0.596 | 0.754 |
| saugeenday-M-12 | 0.334 | 0.334 | 0.324 | 0.379 | 0.296 | 0.293 | 0.415 | 0.405 | 0.278 | 0.342 | 0.326 | 0.322 | 0.445 |
| saugeenday-W-8 | 0.506 | 0.479 | 0.449 | 0.449 | 0.363 | 0.372 | 0.471 | 0.696 | 0.380 | 0.601 | 0.549 | 0.896 | 0.855 |
| solar-10T-48 | 0.658 | 0.532 | 0.475 | 0.504 | 0.511 | 0.498 | 0.575 | 0.785 | 0.545 | 0.804 | 0.860 | 0.870 | 0.860 |
| solar-10T-480 | 0.353 | 0.353 | 0.356 | 0.350 | 0.436 | 0.453 | 0.681 | 0.573 | 0.359 | 0.516 | 0.771 | 4.590 | 0.771 |
| solar-10T-720 | 0.355 | 0.355 | 0.340 | 0.334 | 0.443 | 0.497 | 0.747 | 0.545 | 0.352 | 0.498 | 0.786 | 2.930 | 0.786 |
| solar-D-30 | 0.284 | 0.279 | 0.281 | 0.278 | 0.287 | 0.286 | 0.328 | 0.396 | 0.269 | 0.278 | 0.282 | 0.281 | 0.757 |
| solar-H-48 | 0.322 | 0.322 | 0.317 | 0.316 | 0.298 | 0.303 | 0.337 | 0.468 | 0.358 | 0.406 | 0.628 | 1.080 | 0.628 |
| solar-H-480 | 0.336 | 0.333 | 0.332 | 0.331 | 0.368 | 0.356 | 0.355 | 0.493 | 0.324 | 0.376 | 0.557 | 2.120 | 1.270 |
| solar-H-720 | 0.341 | 0.339 | 0.338 | 0.343 | 0.405 | 0.373 | 0.448 | 0.512 | 0.324 | 0.493 | 0.607 | 1.720 | 1.470 |
| solar-W-8 | 0.227 | 0.183 | 0.136 | 0.147 | 0.133 | 0.136 | 0.162 | 0.531 | 0.124 | 0.171 | 0.152 | 0.139 | 0.236 |
| temperature-rain-with-missing-30 | 0.551 | 0.551 | 0.548 | 0.548 | 0.538 | 0.544 | 0.609 | 0.791 | 0.565 | 0.586 | 0.694 | 0.754 | 1.630 |
| us-births-D-30 | 0.023 | 0.023 | 0.022 | 0.023 | 0.026 | 0.028 | 0.023 | 0.104 | 0.018 | 0.019 | 0.074 | 0.074 | 0.144 |
| us-births-M-12 | 0.021 | 0.021 | 0.019 | 0.020 | 0.019 | 0.016 | 0.013 | 0.036 | 0.013 | 0.011 | 0.010 | 0.012 | 0.017 |
| us-births-W-8 | 0.016 | 0.016 | 0.015 | 0.015 | 0.013 | 0.013 | 0.010 | 0.027 | 0.011 | 0.013 | 0.018 | 0.018 | 0.022 |

