# OpenReview forum: "Test-Time Efficient Pretrained Model Portfolios for Time Series Forecasting"
_ICLR.cc/2026/Conference — ICLR 2026 Poster_

### Official Review · Reviewer_tn7T · 2025-10-27

**Soundness:** 3
**Presentation:** 3
**Contribution:** 2
**Rating:** 6
**Confidence:** 3

**Summary:**

This paper presents Chroma and framework for having portfolio of specialized models originated by fine-tuning a general purpose model for time-series forecasting. At inference time, this frameworks employs a subset of the models based a Greedy Ensemble Weighting algorithm. The paper provides a lot of empirical evaluations ranging from performance and ablation to test-time compute (in terms of FLOPs) and scaling. The results (including statistical analysis and bias-variance tradeoff) suggest the utility of the proposed framework in terms of time saving while maintaining performance.

**Strengths:**

- The paper is strong in terms of empirical contributions to support claims
- I also found the paper strong in terms of reproducibility
- Overall, I think the paper provides insights for the community that could be interesting therefore, I am in favour of acceptance.

**Weaknesses:**

- I think the paper lacks technical novelty because training a source model and then fine-tuning it for different target tasks has been round in transfer learning (particularly domain adaptation) for a long time. Similarly, the studied selection strategies are also known. Therefore, it seems the main contribution of Chroma is the fine-tuning step as opposed to training from scratch  (comparing figure 1 b and c) but that step according to table 2 did not have an impact on the performance as opposed to the ensembling which makes the technical contributions very limited.

- This is not a weakness per se but I think it will be beneficial to show that the presented results are not only specific to T5 architecture

- I think the paper can benefit from more discussion around G.5 results. Particularly to identify for what type of problems this approach works better and where we should train from scratch. Currently, these results are buried in the appendix but a discussion around them can clarify the applicability domain of the presented framework better.

**Questions:**

- How many times each experiment was repeated? I recognize the paper is quite strong in terms of experiments but providing confidence interval or standard error can make the message stronger specially for key experiments

- Did you perform p-value correction for multiple hypothesis testing? if so, what was the employed method?

---

> ### Author Response · Authors · 2025-11-21
> **Comment by Authors - 1**
>
> We thank the reviewer for their insightful comments and the detailed review of our work. We include our comments and answers below.
>
> **Technical novelty.**
>
> We thank the reviewer for their comment and also for acknowledging the technical contribution of our methodology. However, we would like to highlight two respects where we believe the paper has interesting contributions.
>
> First, we do not study domain adaptation in isolation, but in the backdrop of much smaller-sized models combined at test time. In other words, our objective in post-training is not only domain adaptation toward a single domain, but that domain-adapted small models can be combined at test time to give comparable performance to big models.
>
> We also go more in depth on _why_ ensembling pretrained model works at all, and our results on ensembling work both for post-trained and for “from scratch” training. We are hopeful the reviewer can agree that our technical contribution can be framed slightly more broadly.
>
> **T5-architecture.**
>
> We thank the reviewer for this comment and highlighting this limitation. Exactly quantifying the effect of transformer architectures is an interesting direction. However, modern pretrained time series forecasting models share other choices that we think are much more essential compared to the transformer itself: input patching and multi-step ahead direct prediction of quantiles. We believe our method should generalize well as long as these hold true, as they do in virtually all recent competitive pretrained time series models.
>
> We highlight this as a limitation in our work and offer discussion around architecture choices. Please also see our answer to Reviewer 2k41 for encoder-decoder vs. other architectures.
>
> **More discussion around G5 results.**
>
> We agree with the reviewer that this would indeed be a very good contribution. However, from our experiments, we cannot draw a direct conclusion about how tasks and from-scratch vs post-trained models interact. We have updated our draft to include this discussion. We will also work to understand if similar conclusions can be drawn for the conditions when ensembling is better than model selection.

---

> > ### Author Response · Authors · 2025-11-21
> > **Comment by Authors - 2**
> >
> > Regarding the reviewer's questions:
> >
> > > How many times each experiment was repeated? I recognize the paper is quite strong in terms of experiments but providing confidence interval or standard error can make the message stronger specially for key experiments
> >
> > We conduct five independent trials of post-training for our scaling laws and ablation experiments. For these large scale benchmark experiments, we fix the Chroma portfolio to the portfolio that was trained with the smallest seed (i.e., an arbitrary pretrained model, and not one specially picked “release model” in any way).
> >
> > This manner of single-trial benchmark evaluation is standard in pretrained models, firstly due to the sheer computational requirement. Second, this is because the variance that results from the pretraining procedure alone is often small, and the test-time computation introduces no variance (i.e., is practically deterministic). This is consistent with our experience.
> >
> > Finally, we refrained from providing error bars around aggregated benchmark results for similar concerns to other large scale pretrained model evaluation scenarios. Benchmarks aggregate results over different datasets, which cannot be assumed identically distributed. As such, the variation that results from using different datasets in a benchmark dominates the standard error due to independent trials of the same model. We feel this paints a false picture of how models compare.
> >
> > We hope these points clarify our choices on presenting empirical results. We thank the reviewer again for their suggestions and we are looking forward to hearing any further suggestions on this point.
> >
> > > Did you perform p-value correction for multiple hypothesis testing? if so, what was the employed method?
> >
> > We thank the reviewer for raising this important point. We did not employ p-value adjustment in the main table as these are three distinct pre-specified hypotheses, and the trials that supported them were also independent. Looking at the results more carefully, however, we have indeed reported simple ensemble ablations in the same table without p-value adjustments, and these could be thought of as multiple hypothesis testing. However, note that the p-values associated with this analysis (Table 2 and 7) are all numerical zero, so an adjustment would not change our reporting.
> >
> > For Tables 6 and 8, we conducted mixed-effects regression for each subgroup (line of the table) where the metric is the dependent variable and the independent variable is the ablation indicator. The data for these analyses were mutually exclusive. However, via the reviewer’s question we understand the questions were not independent, and as such, a p-value adjustment within each table for Tables 6 and 8 is required. We will update the numbers in these two tables using the Benjamini-Hochberg procedure. As the reviewer can also verify, doing this results in no significant change of conclusions for Table 6. For Table 8, the p-values reported for Portfolio Type/WQL are no longer significant. We thank the reviewer again for the scrutiny, which helped improve our paper.
> >
> > We thank the reviewer for their insightful questions and comments. We would be happy to address any additional questions they might have.

---

> > > ### Comment · Reviewer_tn7T · 2025-11-25
> > > **Concerns addressed**
> > >
> > > I'd like to thank authors for addressing my concerns and their lively discussions in the rebuttal phase. I'll gladly increase my score.

---

### Official Review · Reviewer_2k41 · 2025-10-29

**Soundness:** 3
**Presentation:** 2
**Contribution:** 2
**Rating:** 4
**Confidence:** 4

**Summary:**

The authors proposed chroma to reframes “bigger is better” for time-series FMs by replacing one large monolith with a portfolio of small specialists. Starting from a generalist base model, the authors fine-tune disjoint subsets to create diverse experts, then use model selection or ensembling at inference. Across standard benchmarks, this portfolio matches large models with far fewer parameters and is more compute-efficient than test-time fine-tuning, suggesting a simple, scalable alternative that may generalize beyond forecasting.

**Strengths:**

Strengths:
1. The problem is well motivated and of interest in the community.
2. The empirical study is comprehensive. The authors investigate different portfolio design and the combination methods. One thing I personally like is that, from Table 2, the findings of simply using ensemble cannot help improve performance.
3. Technical contributions are somewhat limited. But I feel the problems is still worth investigated from this perspective.

**Weaknesses:**

Weakness:
1. The proposed ensemble method is a bit hand-wavy. It shows the good performance but doesn't provide insights on why/how it works. Althrough some ablation studies are provided, I still wonder, in general, what kind of ensembling methods can help to improve the performance and any principles behind that. If we have this kind of insights, that would bring this work to a better level.
2. I am not sure if the conclusion can extend to other model architecture. In this paper, it only uses chronos-bolt which is encoder-decoder architeture. But other time series foundation models use decoder-only (e.g., TimesFM) or maksed-encoder (e.g., Moirai). I think that if only encoder-decoder architecture is considered, maybe it is better to adjust some of the claims and the conclusion to limit the model type.
3. For time series foundation models with MoE, there are some other works. For examples, time-moe and moirai-moe. May also consider has some discussions on these in related works.

References

[1] Time-MoE: Billion-Scale Time Series Foundation Models with Mixture of Experts. ICLR 2025

[2] Moirai-MoE: Empowering Time Series Foundation Models with Sparse Mixture of Experts. ICML 2025

**Questions:**

Questions:
1. post-training helps to reduce training cost but I wonder if it reduces the diversity. Because if we train the specialist models from scratch, the specialist models may learn more on the unique pattern in different subsets. Particularly, if we train from scratch, could we get better performance? Or only similar performance but with significantly more computational cost.
2. How to determine the validation sets, especially if we don't have enough data? Additionally, when we only want to make a forecast for a single time series instead of a whole evaluation dataset, how we can do the forecast combination at test-time?
3. I notice that 1K gradient steps seems sufficient for the fine-tuning (i.e., post-training) and further increases this doesn't help. I wonder if this is because of the data amount used in post-training or any other possible reasons?
4. I wonder if the scaling behavior holds if we further increase the model size to 200M (the size of chronos-bolt-base).

---

> ### Author Response · Authors · 2025-11-21
> **Comment by Authors - 1**
>
> We thank the reviewer for their insightful comments and the detailed review of our work. We include our comments and answers below.
>
> **Choice of the ensemble method, insights on why the method works.**
>
> We thank the reviewer for this thoughtful comment.
>
> As of the time of writing of our paper, time series ensemble methods were broadly limited to the simple approaches we explored in the paper. Previous works often found that while simple combination methods provided some improvement, using more complex models to stack predictions often did not help in forecasting. This dilemma, to draw contrast to other applications of statistical learning, was termed the “forecast combination puzzle.” [1] As such, while we agree with the reviewer that the selection of the forecast combination method is heuristic, it is one of the few flexible combination methods that has emerged from previous studies.
>
> We would like to use this opportunity to highlight our bias-variance discussion in the appendix, on why our method works. As the reviewer correctly observed, “simply using ensembles cannot help improve performance.” We agree, and our discussion offers some perspective on this. First, we observe that in the regime where time-series foundation models operate, the bias component of the error dominates the variance component. As such, any benefit of ensembles should come from decreasing bias and not by simple variance reduction. Ensembling methods like simple averaging focus on the latter. In contrast, our framework focuses on 1/ building a diverse portfolio such that one pretrained model will have bias matching the target dataset, and 2/ having a simple but flexible ensemble scheme to select this model at test time. This is how we achieve bias reduction through ensembling.
>
> We believe that this insight might extend to other foundation model domains and guide how model portfolios are formed. We agree that these findings require more emphasis in the main paper and we used the opportunity of 1-more page in the revised version to emphasize this.
>
> [1] Wang, Xiaoqian, et al. "Forecast combinations: An over 50-year review." International Journal of Forecasting 39.4 (2023): 1518-1547.
>
> **Would our findings extend to other architectures?**
>
> The reviewer correctly points out that while our claims are for general pretrained time series models, our experiments only consider one architecture. We believe our results will generalize to other pretrained modeling paradigms where the model directly predicts quantiles across a fixed time horizon, either via encoder-decoder or encoder-only architectures. We also suspect other models such as TimesFM also stand to benefit. Although TimesFM takes a decoder-only approach it incorporates many common choices such as direct-quantile predictions and patching. We would also like to point out, demonstrated benefits within our experiment setup come from creating diversity through training data, and these findings generalize across architecture dimensions explored. This gives us some hope other architectures will also benefit from our method.
>
> We will, however, highlight this as a limitation in our work and offer discussion around architecture choices. We will also contrast with MoE architectures which we agree may not benefit as much (as they could already be learning “specialists” within their architecture). We also agree the two cited are important to our discussion and have included them in our related work section.

---

> > ### Author Response · Authors · 2025-11-21
> > **Comment by Authors - 2**
> >
> > **Post-training vs “from scratch” in specialist training.**
> >
> > We thank the reviewer for this question, which if we understand correctly is the same one answered in our third ablation study (cf. Table 2 and Table 8). In terms of accuracy, we do not find any statistically significant effect on accuracy.
> >
> > Here, we also provide a rework of Table 1 that has evaluations of portfolios of models trained from scratch:
> > >
> >  | Group               | Strategy           | WQL 1m | WQL 2m | WQL 4m | WQL tiny | MASE 1m | MASE 2m | MASE 4m | MASE tiny |
> > |---------------------|--------------------|--------|--------|--------|----------|---------|---------|---------|-----------|
> > | Domain              | Ensemble (Greedy)  | 0.957  | 0.936  | 0.932  | 0.915    | 0.964   | 0.932   | 0.928   | 0.918     |
> > |              | Model Selection    | 0.963  | 0.950  | 0.939  | 0.923    | 0.973   | 0.948   | 0.933   | 0.930     |
> > | Frequency           | Ensemble (Greedy)  | 0.926  | 0.898  | 0.890  | 0.896    | 0.910   | 0.884   | 0.878   | 0.887     |
> > |       | Model Selection    | 0.918  | 0.916  | 0.880  | 0.909    | 0.920   | 0.899   | 0.887   | 0.893     |
> > | Domain (scratch)    | Ensemble (Greedy)  | 0.953  | 0.927  | 0.923  | 0.911    | 0.947   | 0.924   | 0.925   | 0.905     |
> > |     | Model Selection    | 0.987  | 0.928  | 0.943  | 0.919    | 0.968   | 0.936   | 0.942   | 0.919     |
> > | Frequency (scratch) | Ensemble (Greedy)  | 0.907  | 0.905  | 0.893  | 0.910    | 0.906   | 0.895   | 0.882   | 0.880     |
> > |  | Model Selection    | 0.935  | 0.914  | 0.885  | 0.940    | 0.920   | 0.913   | 0.895   | 0.896     |
> > | Generalists         | Ensemble (Greedy)  | 0.987  | 0.951  | 0.939  | 0.919    | 0.974   | 0.944   | 0.931   | 0.928     |
> > |          | Model Selection    | 0.990  | 0.966  | 0.942  | 0.944    | 0.979   | 0.961   | 0.935   | 0.941     |
> > | Single Generalist   | —                  | 1.000  | 0.977  | 0.960  | 0.958    | 1.000   | 0.971   | 0.962   | 0.961     |
> >
> >
> >
> > It can be seen that there is no significant difference, even though the portfolios trained from scratch required 200x more compute for each specialist.
> >
> > We did not include these results in our initial submission because our statistical analysis in Table 2 showed that the difference compared to post-trained portfolios was insignificant. However, we agree that this is an important finding. Therefore, in the revised version, we have provided this extended version of Table 1 in the appendix.
> >
> > **Validation sets and single time series.**
> >
> > Our work uses rolling-window evaluation (also referred to as backtesting, out-of-time folds, etc.) to perform model selection and evaluation. As such, we conceptually think of “enough data” in terms of the length of the provided series rather than a set of different time series. In this light, even with a single time series, one could perform model or ensemble selection by increasing the number of evaluation windows. That is, instead of using a single window for selecting which specialists provide the best forecasts, one could rely on multiple. We have incorporated additional discussion into the manuscript to make this point more clearer.
> >
> > **Number of gradient steps in post-training**
> >
> > We would like to highlight 1K iterations is not an experimental variable we have treated in depth.
> > However we can speculate based on the experience of working with Chroma and Chronos models. Post-training stands in contrast to our main proposition. We combine models to get better models. While post-training may appear like a “combination” of a pretrained model and patterns that may be induced from a subset, we believe after about 1K steps it increasingly reduces almost to only the latter. After this point, as the reviewer comments, success becomes a function of the dataset’s size and richness, and the task’s regularity. We prefer not to state this as a conclusion of our work, as it requires further evidence and more careful experiment design.
> >
> > **Scaling behavior**
> >
> > We believe our main findings will extrapolate into the 200M scale. However, beyond the scales explored in our paper we can posit that the marginal gains from our method will diminish. This is since as model sizes grow, error due to bias decreases faster than variance. As our framework primarily targets bias reduction through forecast combination, we can predict the effect due to just our method being less pronounced.
> >
> > We thank the reviewer again for their comments that led us to think deeper on the aspects of our work we could not fully highlight. We would be happy to address any additional questions they might have.

---

### Official Review · Reviewer_Z6Sb · 2025-10-31

**Soundness:** 3
**Presentation:** 3
**Contribution:** 2
**Rating:** 2
**Confidence:** 4

**Summary:**

The paper investigates whether building portfolios of smaller pretrained models can match or surpass large monolithic time-series foundation models (e.g., Chronos-Bolt) under constrained compute budgets.

**Strengths:**

- Elegant and Compute-Efficient Framework. The proposed portfolio of specialists provides a pragmatic alternative to large models, achieving substantial compute savings. The post-training step is extremely lightweight (only 1K updates), making the approach appealing for practitioners with limited resources.
- Strong Empirical Insights. The work provides valuable empirical findings: ensemble diversity is critical for the forecasting tasks and ensembling yields a favorable accuracy-compute trade-off close to the efficiency frontier.
- Potential for Broader Impact. By reframing scaling from “bigger models” to “smarter test-time computation,” the study could inform future directions in efficient time-series foundation modeling and resource-adaptive forecasting systems.

**Weaknesses:**

1. **Validation data at test time is impractical and misaligned with typical deployment.** The method assumes the availability of non-trivial validation data at test time to select or weight experts. In realistic forecasting deployments, only inputs (and possibly a short warm-up context) are available. Labeled validation slices are rare or prohibitively small. This creates a distributional and resource gap between the paper’s evaluation protocol and common practice.
2. **Zero-shot and OOD forecasting ability are not validated.** Although the paper positions the approach against “zero-shot” foundation models, the core mechanism, model selection or greedy ensembling using validation losses, consumes task labels and thus departs from zero-shot forecasting. **The zero-shot experiments are highly recommended, especially when the test data have a significant domain shift with the validation data.**
3. **Greedy ensemble selection risks overfitting to the validation window.** The ensemble is built by iteratively adding experts to minimize validation error. Repeated selection against a small, local window is prone to selection bias and variance inflation, particularly with correlated experts. We may need further experiments to show that this method has resistance against overfitting
4. **Scaling-law claims rest on a narrow 1–9M parameter band.** Figure 3 supports “comparable scaling to single models” only in a small-model regime. Extrapolating this trend to ≥100M/1B parameters is not justified without additional anchor points, thus being unconving.
5. **Expert construction is largely manual and metadata-dependent.** Experts are partitioned by frequency/sector categories, and those categories are highly dependent on the manual design. Hence, it is unclear how this manually designed expert construction performs on the unseen domain/tasks.
6. **Experimental gains are modest and sometimes negative.** Across benchmarks, the ensemble often only slightly outperforms the best single expert. On Chronos, the 4M-parameter ensemble even underperforms the top single model.

**Questions:**

See the weaknesses

---

> ### Author Response · Authors · 2025-11-21
>
> We thank the reviewer for their review of our work. We believe that there might be significant confusions in their conclusions. We hope our comments below can address these:
>
> **Validation data.**
>
> We thank the reviewer for this comment, however we believe it stems from a significant confusion. Our method only requires one horizon-length window in context to perform model selection / forecast combination, and our models are otherwise pretrained. This is significantly more data-efficient than common forecasting practice where the same data should also be used to fit forecasting models. As it stands, this comment appears to be debating whether _any_ context data is available at test-time–which the literature assumes the answer is yes. We hope that this clarifies a possible confusion.
>
> **Zero-shot and OOD forecasting.**
>
> We want to clarify that all our benchmarks are zero-shot. In other words, all of our evaluation datasets were excluded from model pretraining. We do not use task _labels_, but the task _context_ during test-time to perform model selection. To our understanding, the argument the review is making can be reduced to: “any forecasting method that looks at the past is not zero-shot, because they have looked at the past to predict the future.” Again, we believe that this is not a counter-argument to our work but a refutation of commonly accepted nomenclature in literature.
>
> **Greedy ensembles and overfitting.**
>
> The greedy ensemble used in our work assigns a single weight to a constituent model, _across all items_ and _across the prediction horizon_. In other words, the number of weights in the learned ensemble itself is <10. We fail to see how overfitting could result in this scenario. Moreover the central point of our paper is – because pretrained models have high bias, model selection and ensembles result in dramatic bias reduction compared to any variance inflation–even if we learned more complex ensemble models and this was indeed a significant effect. These points are evidenced in our experiments which cover a large variety of tasks with small, medium and large datasets.
>
> **Scaling laws.**
>
> We agree with the reviewer that this is indeed a limitation: scaling laws studies imply extrapolation (also please refer to our answer to Reviewer 2k41). But once more, this debates the framework of scaling laws rather than our paper in particular. Like all scaling laws studies, our study covers the range of parameter sizes that are interesting in the backdrop of time series forecasting. Modern monolithic pretrained time series forecasting models have 100-200M parameters, and we explore small specialists that are an order of magnitude smaller.
>
> **Expert construction.**
>
> We agree with the notion that our expert construction is heuristic and metadata-dependent. However, we do not argue in the paper that this construction is in any way optimal, but just that such constructions exist. As such, being able to manually construct portfolios through metadata commonly available in time series benchmarks and databases, in a simple and straightforward manner, speaks for and not against the main finding.
>
> We would also like to kindly ask for clarification on the “unseen” domains and frequencies the review hints at. Our domain and frequency categorization are both collectively exhaustive by design. Our domain categorization includes a residual “various” category, and our frequency categorization features “sub-hourly” and “supra-monthly” categories. As such, the knowledge gap the comment hints at is already part of our experiment setup.
>
> **Experimental gains.**
>
> We would appreciate it if the reviewer could kindly state to which benchmark they are referring to. In all experiments, model portfolios outperform single generalist models by a large margin. The Chronos-BM2 and GIFT-eval benchmark studies compare our models to previous work which are 1-2 orders of magnitude larger in size, but always outperform comparably sized models from previous works.

---

> ### Comment · Reviewer_Z6Sb · 2025-11-23
>
> Thanks for the detailed response. Most of my concerns have been well addressed, and I really appreciate the authors' efforts on it. But I still have some question about the model selection/combination since I seem to have a misunderstanding on this part.
>
> First, about the validation set, I am curious about **the size of the validation set**, since the author claims that the validation set is just one horizon-length window. Moreover, suppose we have a history window $X$ and we want to predict the future $Y$, do the author indicate that **the input history window $X$ can serve as the validation set to find the best model or find the best weight for model combination**?
>
> Second, about the zero-shot ability, I am not indicating that a model does not have zero-shot ability if they use the past to predict the future. My concern is still about the model selection/selection. (1) If the answer to the first question is yes (the input history window can be used as the validation set), then I agree that this method has a strong zero-shot ability; (2) If the answer is no (the method requires a large validation set to find the optimal weight on a certain dataset), then I am curious about the performance when using the optimal weight on one dataset (such as Chronos Benchmark) to another (such as GIFT-Eval).

---

> ### Author Response · Authors · 2025-11-23
>
> We thank the reviewer for their prompt response. Regarding their questions:
>
> **The size of the validation set:**
>
>
> In general, our work uses rolling window evaluations based on backtesting.
>
> Let's break this down  more formally. A time series task can be thought of as a collection of time series indexed by $i \in \{1, 2, \cdots, N\}$, $(y_{i, 1:T}, y_{i, T+1:T+H})$, where the objective is to forecast $y_{i, T+1:T+H}$ given $y_{i, 1:T}$. In our terminology, we refer to the former as the forecast horizon (in our response, the _task label_) and the latter as the _context_. Our model selection approach further splits the context window into $(y_{1:T-H}, y_{T-H+1:T})$. We refer now to $y_{T-H+1:T}$ which is part of the original task _context_, as the validation window. We use our models to predict this window using $y_{1:T-H}$ and perform model selection / ensembling. The size of the validation set is therefore a function of number of time series $N$. In single-time-series tasks, it is a single validation window. In larger tasks, it is in the thousands. One could extend this framework to rely on multiple windows for backtesting. The setup we explore is therefore the most conservative.
>
> We have added additional discussion to our revised manuscript clarifying this point.
>
> **Zero-shot ability**
>
> > Do the author indicate that the input history window can serve as the validation set to find the best model or find the best weight for model combination?
>
> Yes, this is correct. Time series forecasting tasks naturally contain past values of the input time series as a context, and we use a part of this context for validation. Therefore, our method does not require any additional data compared to other approaches. As the reviewer stated in their response (1), this makes our methodology zero-shot.
>
> We hope these answers clarify any confusion. We thank the reviewer again and would be happy to address any additional questions they might have.

---

> > ### Comment · Reviewer_Z6Sb · 2025-11-23
> >
> > Thanks for the authors' explanation. I will increase the score to at least 4.
> >
> > After carefully reading the paper, I still have some concerns about the experimental setting. Following the authors' notation, could the author provide the precise number of the history horizon $T$, the validation window length $H$, and the forecast horizon $H^\prime$? I would really appreciate it if the author could provide the detailed experimental setting for both Chronos Benchmark II and GIFT-Eval, since I cannot find the correspondinging explanation in the paper.

---

> ### Author Response · Authors · 2025-11-24
>
> We thank the reviewer for their prompt response, and this question.
>
> The specific forecast horizons used during evaluation are fixed by the benchmarks, and they differ among different datasets. We refer the reviewer to the respective papers for this detail: GIFT-eval [1, Table 13], and Chronos BM2 [2, Table 3, as "Zero-shot evalution"], denoted "Prediction Length." In our notation the validation prediction lengths and the task prediction lengths are the same $H = H'$.
>
> We fix $T$ to a _maximum_ of Chroma models' context lengths which is 2048. However, the specific actual number $T$ is also a property of the benchmarks, and varies among datasets as well as among individual time series within each dataset. The reviewer can refer to the same tables above for these statistics.
>
> We are also happy to address any other concerns or questions on the details of the experiment setup.
>
> [1] Aksu, Taha, et al. "Gift-eval: A benchmark for general time series forecasting model evaluation." arXiv preprint arXiv:2410.10393 (2024).
>
> [2] Ansari, Abdul Fatir, et al. "Chronos: Learning the language of time series." arXiv preprint arXiv:2403.07815 (2024).

---

> > ### Comment · Reviewer_Z6Sb · 2025-11-24
> >
> > Thanks for the detailed response. My concerns have been well solved.

---

### Official Review · Reviewer_QGqS · 2025-11-02

**Soundness:** 3
**Presentation:** 3
**Contribution:** 4
**Rating:** 8
**Confidence:** 4

**Summary:**

This paper investigates fine-tuning/post-training of time series foundation models. Rather than using pre-trained TSFMs for zero-shot forecasting directly, they propose to specialize a base model into different groups (can be flexibly defined, but the paper experimented with domains and frequency groupings). Given a group of models, a single forecast can be obtained either by model selection, or model ensembling. Experiments are done to identify the best approach. Results show that an ensemble of small models are on par with much larger models.

**Strengths:**

This is a very interesting investigation into the capabilities of time series foundation models. Paper is well written and well positioned in the literature. The paper proposes an innovative method to utilize pre trained time series models, and performs extensive experiments, yielding useful insights.

**Weaknesses:**

* Scaling results are not strong. There are likely too few data points across the scaling axis to reliably trust the results. For frequency specialists, 4m models are consistently stronger than tiny (9m) models. There is no clear indication we should expect the regression to extrapolate. This is also reflected in the main results figure, leading to some confusion when one is reading about scaling behavior but in the main results, 4m is clearly stronger than 9m.
* It is unclear what "active parameters" refers to in this setting. From my reading I understand active parameters for an ensemble of 4m models to be 4m. However, I would think that for model ensembles, the number of active parameters should be ensemble size * model size
* Omits more recent models with much stronger results, e.g. Toto, Tirex, TimesFM 2.5

**Questions:**

* if model selection/ensembling is selected using the first evaluation window, how are the metrics for this first evaluation window obtained without using the ground truth values? Shouldn't the window before the first evaluation window be used for selecting the model combination method instead?
* Is the same model combination selection method also used for other experiments, e.g. scaling plot?

---

> ### Author Response · Authors · 2025-11-21
>
> We thank the reviewer for their insightful comments and the detailed review of our work. We include our comments and answers below.
>
> **Scaling results**
>
> We agree with all of the reviewer’s points. While we believe these scaling laws can extrapolate, this may well be contingent on more data being provided. Moreover, our framework hits the data limit earlier than generalist training (for which scaling curves are better-behaved). One can imagine this is because certain frequency or domain buckets hit the data limit faster.
>
> We believe our scaling observations cover and could potentially extrapolate in the region that is of interest to us – in the <20M parameter range. Recent pretrained time series forecasting models are ~200M (e.g., TimesFM 2.5) parameters in size. As our main contribution here is to maintain portfolios of expert much smaller models, we are not mainly interested in extrapolating scaling behavior we find to this range.
>
> **Active parameters**
>
> Thank you for this question. Active parameters of model portfolios consisting of 4m models are “4m” if the underlying methodology is model selection. In case the methodology is ensembling, it would depend on the ensemble size; and would be equal to 4m if the best ensemble consists of one model, 8m if two models, and so on.
> We now understand how this point is not very clear, and we revised that part as follows:
>
> > “We find that, despite having as little as 4M active parameters at test time, the Chroma portfolio *under best model selection* performs comparably to much larger monoliths such as Moirai-1.1 Large (311M parameters) , TimesFM-2.0 (500M parameters) or Chronos-Bolt Base (205M parameters).”
>
> **Omitting more recent models (Toto, Tirex, TimesFM 2.5)**
>
> We acknowledge that there are some recent models performing better on these benchmarks. However, we hope the reviewer agrees that we are not claiming to outperform the SoTA models on these benchmarks, but that our method performs well on them using a creative way to leverage small models. As these benchmarks are quite popular and widely available in a live leaderboard, we believe not repeating these results here is not a significant misrepresentation. We are happy to add some of these models, however, if the reviewer feels it will add an interesting dimension to the discussion of this paper.
>
> Regarding the reviewer's questions:
>
> **Rolling window evaluation**
>
> We are very grateful to the reviewer for identifying this clerical error in the paper. To clarify: in GIFT-eval rolling window evaluation tasks, the task is split into multiple windows that are from the same time series. The paper is trying to highlight that it took the hard path for computational efficiency: the ensemble was fixed in the _training window_ of the _first rolling window_, and was not “refit” in subsequent windows. We can confirm that no test leakage occurred, as you may have naturally suspected from this wording.
>
> **Model combination selection**
>
> Yes, the same model combination selection is used for all plots. We now have explicitly state in the main paper that we use the same methods throughout all experiments.
>
> We thank the reviewer again for their insightful questions and comments. We would be happy to address any additional questions they might have.

---

### Author Response · Authors · 2025-11-21
**Summary of main revisions**

We thank all the reviewers for their time and for their comments and suggestions that have improved our work.

In this general comment, we would like to highlight the main changes in the revised manuscript:

1. More discussion of our bias–variance decomposition-based explanation of the underlying conclusions in the main text.

2. More explicit discussion of the model architecture used in our experiments and providing discussion on comparison to other architectures.

3. Added Table 9, which is an extension of Table 1 that also includes model portfolios trained from scratch.

4. Adjusted p-values in Tables 6 and 8 for multiple hypothesis testing.

We have also incorporated the other comments by the reviewers. The changes are highlighted in blue.

We thank the reviewers again and would be happy to address any further questions they might have.

---

### Public Comment · ~Quang_Truong1 · 2026-03-24
**Questions regarding the extra training data**

Dear authors,

Thank you very much for your work. I have 2 questions regarding the extra datasets that you listed in Table 4.

1. For the variants without missing data, I observe that the number of observations still remains the same for some datasets (e.g. monash/covid_mobility_without_missing_values), but drops for monash/bitcoin_without_missing_value (but the math doesn't exactly match since the ratio of after/before is only around 8.6% vs. 9.8% reported in the paper). I wonder if you performed any imputation for NaN values? And why was monash/bitcoin_without_missing_value treated differently, as well as why is there a slight mismatch in ratio?
2. For other datasets in Monash and UCI repository, do you use the entire sequence for training? I'd also appreciate if you can provide public URLs to the exact datasets that you used in the paper for better reproducibility.

Again, thank you very much for your work!

---

### Meta-Review · Area_Chair_hpnr · 2026-01-05

**Summary:**

This well-written paper has been assessed by four knowledgeable reviewers whose initial scores put it at the borderline (one straight reject, one marginal reject, one marginal accept, and one straight accept scores).  The initial impression of the reviewers was mixed. They acknowledged good quality of presentation and that it tackles an important problem, provides comprehensive experiments with useful insights and strong reproducibility. However, they were concerned with limited technical novelty (the key  concepts relied upon such as fine-tuning or ensembling are not new), weak and inconsistent scaling. They were worried that claims may not generalize beyond the tested architecture, and the omission of recent stronger models and deeper theoretical explanations would further limit the impact. However, the authors vigorously engaged the reviewers in discussions and provided detailed rebuttals which addressed most of the pressing comments. This paper, assuming the authors update its contents to reflect the results of those discussions, is fit for inclusion in ICLR.

**Reviewer Concerns:**

Most of them have been quite effectively addressed, I see nothing substantial left over that would preclude the acceptance.

**Reviewer Scores:**

Some of the reviewers increased their scores actually. The one with the weakest score (2) had been effectively persuaded in the discussion with authors, and it looked like they should be substantially improving their score.

---

### Decision · Program_Chairs · 2026-01-26

Accept (Poster)